# Numerical Analysis of the Effect of Heterogeneity on CO<sub>2</sub> Dissolution Enhanced by Gravity Driven Convection

Yufei Wang<sup>1,2,3,4</sup>, Daniel Fernàndez-Garcia<sup>1,2</sup>, and Maarten W. Saaltink<sup>1,2</sup>

**Correspondence:** Yufei Wang (yufei.wang@pku.edu.cn), Daniel Fernàndez-Garcia (daniel.fernandez.g@upc.edu), and Maarten W. Saaltink (maarten.saaltink@upc.edu)

Abstract. Dissolution trapping of  $\mathrm{CO}_2$  in brine can mitigate the risk of supercritical  $\mathrm{CO}_2$  leakage during long-term Geological Carbon Sequestration (GCS). The dissolution of overlying supercritical  $\mathrm{CO}_2$  into brine increases the density of brine in its upper portion, which causes Gravity-Driven Convection (GDC) and thus significantly increases the rate of  $\mathrm{CO}_2$  dissolution. To date, most studies on GDC-driven dissolution are based on homogeneous media and only few studies exist on the effect of heterogeneity on GDC-driven dissolution. Here, we study the effect of heterogeneity and anisotropy on GDC-driven dissolution rate by using numerical simulations with randomly obtained permeability fields. Dissolution rates calculated by these simulations are related to properties of the permeability field by using least-squares regression. We obtained two empirical formulas for predicting the asymptotic GDC-driven dissolution rate. In the first formula the dissolution rate is almost linearly proportional to the dimensionless equivalent vertical permeability. In the second one the dissolution rate is linearly proportional to a dimensionless vertical finger-tip velocity. This indicates that the GDC-driven dissolution can be predicted using either the equivalent vertical permeability or the vertical finger-tip velocity. Furthermore, both formulas demonstrate that higher permeability anisotropy results in lower dissolution rates, suggesting that pronounced horizontal stratification can inhibit the dissolution of  $\mathrm{CO}_2$ .

## 1 Introduction

Geological Carbon Sequestration (GCS) has proved to be a feasible and necessary approach to mitigate CO<sub>2</sub> emissions (Vilarrasa et al., 2013, 2014; European Commission, 2014; Vilarrasa and Carrera, 2015; Tutolo et al., 2014, 2015b, a; Celia et al., 2015). GCS consists of sequestering CO<sub>2</sub>, separated from other gases at large point sources (e.g., thermal power plants), into geological brine formations (Vilarrasa et al., 2010; Brainard, 2018; Matacic, 2018). The injected CO<sub>2</sub>, usually in supercritical state under reservoir condition, is expected to flow over the denser brine during the process of injection, and continuously move upwards and accumulate beneath a low permeability caprock after injection. Because the less dense CO<sub>2</sub> is immiscible but can dissolve in the resident brine, the injected CO<sub>2</sub> is sequestered by four mechanisms: (1) by being sealed under the low per-

<sup>&</sup>lt;sup>1</sup>Dept. of Civil and Environmental Engineering. Universitat Politècnica de Catalunya, Jordi Girona 1-3, 08034 Barcelona, Spain

<sup>&</sup>lt;sup>2</sup>Associated Unit: Hydrogeology Group (UPC-CSIC), Barcelona, Spain

<sup>&</sup>lt;sup>3</sup>Dept. of Applied Mathematics. IFP Energies nouvelles, 1 et 4 Avenue du Bois-Préau, 92825 Rueil-Malmaison, France

<sup>&</sup>lt;sup>4</sup>School of Earth and Space Sciences, Peking University, No.5 Yiheyuan Road Haidian District, 10087 Beijing, China.

meability caprock (hydrodynamic trapping), (2) by being trapped by capillarity (residual trapping), (3) by dissolving into the brine at the  $\rm CO_2$ -brine interface (dissolution trapping) and (4) by reacting with rock formation (mineral trapping) (Kumar et al., 2005; Riaz et al., 2006; Bachu et al., 2007; Gasda et al., 2011, 2012; Macminn and Juanes, 2013; Newell and Shariatipour, 2016; Soltanian et al., 2017; English et al., 2024; Medici et al., 2024; Saló-Salgado et al., 2024). Hydrodynamic trapping is the fastest mechanism but is unstable, because the relatively light  $\rm CO_2$  can escape from the reservoir in case of seismic activity, faults or failing wellbore casings (Vilarrasa, 2012). Mineral trapping is most stable but slow and usually negligible (Baines and Worden, 2004; Saaltink et al., 2013; Sathaye et al., 2014). During the later stage of injection, the safety of the GCS is governed by the dissolution trapping, which is not affected by the non-integrity (e.g., faults) of the formation (Strandli and Benson, 2013; Nicot, 2008). Therefore, the dissolution of  $\rm CO_2$  into brine is an important indicator to evaluate the safety of long-term  $\rm CO_2$  sequestration (Anbar and Akin, 2011; Xiao et al., 2019).

The dissolution process is enhanced by Gravity-Driven Convection (GDC) (Lindeberg and Wessel-Berg, 1997; Lindeberg and Bergmo, 2003; Lindeberg and Wessel-Berg, 2011; Tsai et al., 2013; Sathaye et al., 2014). Dissolution of the overlying CO<sub>2</sub> into the brine increases the brine density in the upper portion of the saline formation creating an unstable situation with more dense on top of less dense brine. This causes nonuniform GDC in the brine phase and enhances the downward movement of CO<sub>2</sub>-saturated brine (Weir et al., 1996; Vella and Hupper, 2006; Pritchard, 2007; Pruess and Zhang, 2008). Moreover, the nonuniform downward flux of CO<sub>2</sub>-saturated brine is accompanied by an upward flux of CO<sub>2</sub>-unsaturated brine, increasing the contact between CO<sub>2</sub>-rich phase and unsaturated brine, which further accelerates CO<sub>2</sub> dissolution (Pruess, 2005; Gilfillan et al., 2009; Elenius and Gasda, 2021). In contrast to pure molecular diffusion, which quickly fades away as the concentration profile reaches a Gaussian distribution, the GDC can fuel the vertical mass exchange at a high constant rate for a certain long time until the whole domain is close to full CO<sub>2</sub> saturation (Pau et al., 2010; Slim, 2014). Because the dissolution rate enhanced by GDC can significantly reduce the amount of supercritical CO<sub>2</sub>, thus reducing the possibility of leakage, a good understanding of this dissolution is essential (Elenius et al., 2015).

Currently, a large quantity of studies exists on GDC-driven dissolution, ranging from theoretical analysis (e.g., Elenius et al., 2012; Emami-Meybodi, 2017), laboratory experiments (e.g., Neufeld et al., 2010; Rasmusson et al., 2017; Salibindla et al., 2018; Mahmoodpour et al., 2019), numerical simulations (e.g., Chen et al., 2013; Elenius et al., 2015; Soltanian et al., 2016; Amooie et al., 2018) to field observations (e.g., Sathaye et al., 2014). It is found that GDC depends to a large extent on properties of the formation (e.g., the permeability and porosity) and of the brine phase (e.g., the relation between brine density and concentration of dissolved CO<sub>2</sub>, viscosity and molecular diffusion coefficient) (Hassanzadeh et al., 2007, 2008; Neufeld et al., 2010; Elenius and Johannsen, 2012; Emami-Meybodi and Hassanzadeh, 2015). For instance, researchers find a linear relation between the dissolution rate and the intrinsic permeability for isotropic homogeneous media (Rasmusson et al., 2015). However, most of the current researches and conclusions are limited to (isotropic or anisotropic) homogeneous fields (e.g. Ennis-King and Paterson, 2005; Pruess and Nordbotten, 2011; Cheng et al., 2012; De Paoli et al., 2017; Ranganathan et al., 2012; Taheri et al., 2012).

Studies on heterogeneous fields are usually limited to qualitative analysis of the effect of heterogeneity (Agartan et al., 2015; Lin et al., 2016; Soltanian et al., 2017; Kim et al., 2019; Yan et al., 2019; Wang et al., 2021; Elgahawy and Azaiez, 2021).

Heterogeneity in permeability plays an important role in onset, growth, maintenance and decay of the GDC and the dissolution rate (Schincariol et al., 1997; Simmons et al., 2001). Frykman and Wessel-Berg (2014) show that we may overestimate the effects of instability in a real heterogeneous field simply by conducting numerical simulations using a homogeneous field of equivalent permeability. Prasad and Simmons (2003) find that while it can trigger early instability, heterogeneity can reduce the instability by dissipating the density fingers through disordered high permeability pathways at a later stage.

Although several researches attempt to quantitatively analyze the relation between GDC and field properties for heterogeneous media, only a few offer explicit formulas between the GDC-driven dissolution rates and permeability heterogeneity. For instance, Farajzadeh et al. (2011) and Kong and Saar (2013) conducted numerical simulations of GDC in isotropic heterogeneous media, but did not offer a quantitative formula between  $CO_2$  dissolution rates and heterogeneity measures. Therefore, conclusions from these researches cannot be directly applied to estimate the dissolution rate in real reservoirs. Several researches offer quantitative formulas for predicting dissolution rates but only consider simplified binary heterogeneous media (Elenius and Gasda, 2013; Green and Ennis-King, 2014; Elgahawy and Azaiez, 2021) or homogeneous media with anisotropic permeability (Erfani et al., 2022). In these researches, different results are obtained regarding the effect of permeability anisotropy on the dissolution rate. While Elenius and Gasda (2013) claimed that the dissolution rate can be estimated without incorporating the impact of anisotropy, Green and Ennis-King (2014) and Elgahawy and Azaiez (2021), respectively, showed positive and negative impacts of horizontal to vertical permeability ratio on the dissolution rate.

Overall, we have a solid understanding of the GDC-driven dissolution process in isotropic homogeneous media, but the GDC-driven dissolution in heterogeneous media needs further study. Especially, we need to quantitatively clarify the impact of the anisotropy ratio on the effective dissolution rate. Moreover, the current predictors are all based on the (equivalent) permeability, and it remains unclear whether we can predict the dissolution rate based on other formation properties or field observations, such as the finger-tip velocity.

Therefore, the objective of this work is twofold. (i) To quantitatively analyze the effect of permeability heterogeneity and anisotropy on the GDC-driven dissolution rate in a wide range of (isotropic and anisotropic) heterogeneous fields with varying degrees of heterogeneity and anisotropy. (ii) To investigate whether the dissolution rate can be predicted based on the finger-tip velocity. We do this in two steps. First, performing numerical simulations over a large number of heterogeneous fields of different permeability distributions. Numerical simulations are carried out by a finite-difference numerical program developed by Wang (2022). Permeability fields are generated with the sequential Gaussian simulation method implemented in the SGSIM code (Journel and Huijbregts, 1976). Second, the results of the simulations are analyzed to find relations among the GDC-driven dissolution rate, permeability heterogeneity, anisotropy and finger-tip velocity, and we compare our results against those given in literature. In this step, ordinary-least-squares linear regressions are used. The conclusions from this work may hold significant relevance for other gravity-driven convection processes, where density differences play a crucial role. These processes include contaminant migration, geothermal exploitation, saltwater intrusion, and mineral precipitation/dissolution (Berhanu et al., 2021; Sanz et al., 2022; Guevara Morel and Graf, 2023; Fang et al., 2024; Liyanage et al., 2024).

The rest of this paper is organized as follows. We give a concise description of GDC in Section 2. We review existing formulas for GDC driven dissolution rates in section 3. Section 4 describes the computational approach. Section 5 gives the results and discussions. Major conclusions are listed in section 6.

## 2 Gravity-Driven Convection

100

105

During the later stage of injection, the injected less dense  $CO_2$ -rich phase floats over the brine, as shown in Figure 1. This  $CO_2$  moves upwards until it reaches a sealing caprock that traps the  $CO_2$ . The trapped  $CO_2$  remains stored in the reservoir over extended periods, contributing to long-term carbon storage. Far from the injection site, a long quasi horizontal interface forms between the brine phase and the  $CO_2$ -rich phase. This interface represents a boundary of constant  $CO_2$  concentration for the underlying brine, from which the overlying supercritical  $CO_2$  dissolves into the brine. Importantly, brine with high  $CO_2$  concentration has  $\sim 1.0\%$  higher density than the underlying brine of very low  $CO_2$  concentration, triggering Gravity-Driven Convection (GDC) that remarkably increases the  $CO_2$  dissolution rate (see Figure 1). GDC continues until the whole reservoir is fully saturated with  $CO_2$ , which may take a long time. The dissolution process can be divided into 4 regimes (Slim, 2014; Tsinober et al., 2022). The first regime is called 'diffusive regime', in which the mass flux is controlled by Fickian diffusion that fades away exponentially, as can be seen from Figure 1. After this, GDC starts to develop and dissolution switches from diffusion controlled to convection controlled. During this regime, which is called 'flux-growth regime', the dissolution rate starts to increase. Once the GDC is fully developed, the dissolution rate reaches a quasi constant value for a relatively long time. This regime is known as 'constant-flux regime'. The last regime (not shown in Figure 1) is referred to as 'shut-down regime' or flux decay regime. In this regime, the dissolution rate decreases as the reservoir becomes fully saturated with  $CO_2$  (Elenius et al., 2015).

In this work, we focus on the dissolution rate taking place in the 'constant-flux regime'. This regime controls the overall behavior of the dissolution efficiency during the geological carbon sequestration (Tsinober et al., 2022), as the diffusive regime typically lasts for a very short duration (see the supplementary information). In this 'constant-flux regime', the dissolution rate can be approximated by an asymptotic dissolution rate, as shown in Figure 1.

## 3 Review of Formulas for GDC-Driven Dissolution Rates

In isotropic homogeneous porous media, the asymptotic dissolution rate  $[kg \cdot m^{-2} \cdot s^{-1}]$  can be written as (e.g., Hesse, 2008; Pau et al., 2010; Slim, 2014),

$$F_{\infty} = \gamma X_0^C \rho_0 \frac{\Delta \rho g \kappa}{\overline{u}},\tag{1}$$

where  $\gamma$  [-] is a scaling coefficient,  $X_0^C$  [kg·kg<sup>-1</sup>] and  $\rho_0$  [kg·m<sup>-3</sup>] respectively represent the  $CO_2$  mass fraction in brine and the brine density at the interface of brine and  $CO_2$ -rich phase,  $\Delta \rho$  [kg·m<sup>-3</sup>] is the density increase when aqueous  $CO_2$  mass

Figure 1. (a) Sketch of the appearance of unstable fingers at the interface of brine and  $CO_2$ -rich phase, and (b) enhanced dissolution rate due to vertical mass exchange driven by unstable fingers (I: diffusive regime; II: flux-growth regime; and III: constant-flux regime).

**Table 1.** Published data for the scaling coefficient  $\gamma$  in isotropic homogeneous field.

| $\gamma$             | Top boundary         | Method    | Reference                    |
|----------------------|----------------------|-----------|------------------------------|
| 0.015-0.017          | diff. $only^{[a]}$   | Num.      | Pruess and Zhang (2008)      |
| 0.017                | diff. only           | Num.      | Hesse (2008)                 |
| 0.017-0.018          | diff. only           | Num.      | Pau et al. (2010)            |
| $0.12Ra^{-0.16[e]}$  | -                    | Num.,Exp. | Neufeld et al. (2010)        |
| $0.045 Ra^{-0.24}$   | Permeable            | Exp.      | Backhaus et al. (2011)       |
| 0.017                | diff. only           | Num       | Cheng et al. (2012)          |
| 0.02                 | diff. only           | Num.      | Elenius and Johannsen (2012) |
| 0.02                 | diff. only           | Num.      | Elenius et al. (2012)        |
| 0.075                | $\mathrm{CTZ}^{[b]}$ | Num.      | Elenius et al. (2012)        |
| $0.037 Ra^{-0.16}$   | Permeable            | Exp.      | Tsai et al. (2013)           |
| $0.0794 Ra^{-0.168}$ | diff. only           | Theor.    | Farajzadeh et al. (2013)     |
| 0.017                | diff. only           | Num.      | Slim (2014)                  |
| 0.025                | $(0.2\kappa^{[c]})$  | Num.      | Slim (2014)                  |
| 0.044                | $(0.6\kappa^{[d]})$  | Num.      | Slim (2014)                  |
| 0.065                | CTZ                  | Num.      | Martinez and Hesse (2016)    |
| 0.018-0.019          | diff. only           | Num.      | Martinez and Hesse (2016)    |
| $Ra^{-0.2154}, 0.06$ | Permeable            | Exp.      | Rasmusson et al. (2017)      |
| 0.09                 | Permeable            | Num.      | This study                   |

<sup>[</sup>a] The top boundary only allows mass to go through the boundary via diffusion.

fraction increases from the initial very small background value to  $X_0^C$ , g [m·s<sup>-2</sup>] is the gravitational acceleration,  $\kappa$  [m<sup>2</sup>] is the intrinsic permeability, and  $\overline{\mu}$  [pa·s] is the mean viscosity of brine. Different researches have obtained different values for the scaling coefficient  $\gamma$  that varies from  $\sim 0.015$  to  $\sim 0.075$ , and a summary of  $\gamma$  is given in Table 1.

In heterogeneous porous media, the intrinsic permeability can vary by several orders of magnitude (Elenius and Johannsen, 2012; Elenius et al., 2012). Therefore, it is important to understand the effect of permeability heterogeneity on the GDC dissolution. Although there exists a large amount of studies on GDC dissolution in heterogeneous porous media (e.g. Green and

 $<sup>^{[</sup>b]}$  The top boundary is capillary transition zone.

 $<sup>^{[</sup>c]}$  The permeability of the top boundary is 0.2 the permeability of the domain.

 $<sup>^{[</sup>d]}$  The permeability of the top boundary is 0.6 the permeability of the domain.

 $<sup>^{[</sup>e]}$  Ra is Rayleigh number (cf. (Neufeld et al., 2010)).

**Table 2.** Summary of the  $\eta$  for equation (3) in the literature.

|                      | Hatanaganaite Tena        | Number of simulations | Source                      |  |
|----------------------|---------------------------|-----------------------|-----------------------------|--|
| η Heterogeneity Type |                           | Number of simulations | Source                      |  |
| 0                    | Binary field <sup>a</sup> | 34                    | Elenius and Gasda (2013)    |  |
| 0.5                  | Binary field              | 24                    | Green and Ennis-King (2014) |  |
| 0.5                  | Homogeneous               | 3                     | De Paoli et al. (2017)      |  |
| 0.21                 | Homogeneous               | 49                    | Erfani et al. (2022)        |  |

<sup>&</sup>lt;sup>a</sup> Binary field means homogeneous field with horizontal barriers.

Ennis-King, 2010; Elenius and Gasda, 2013; Green and Ennis-King, 2014; Taheri et al., 2018; Mahyapour et al., 2022), only a few researches offer concise formulas for predicting the dissolution rate. By conducting numerical simulations in homogeneous medium with embedded horizontal barriers, Elenius and Gasda (2013) proposed that the asymptotic dissolution rate can be estimated by

$$F_{\infty} = \gamma X_0^C \rho_0 \frac{\Delta \rho g \kappa_g}{\overline{\mu}} \left( \frac{\kappa_z^e}{\kappa_g} \right), \tag{2}$$

where  $\kappa_z^e$  is the equivalent vertical intrinsic permeability of the heterogeneous medium, and  $\kappa_g$  is the geometric mean of the permeability field. Essentially, these authors proposed to replace the intrinsic permeability by its equivalent quantity in heterogeneous media. Subsequent studies analyze whether and how the anisotropic effect of the permeability distribution affects the GDC, but the results are inconsistent (Xu et al., 2006; Green and Ennis-King, 2010; Cheng et al., 2012; Chen et al., 2013; Green and Ennis-King, 2014; Kim, 2014; Soltanian et al., 2017; Elgahawy and Azaiez, 2021). Inspired by the result for the isotropic heterogeneous field, researchers propose that the dissolution rate in the anisotropic heterogeneous field can be expressed by

$$F_{\infty} = \gamma X_0^C \rho_0 \frac{\Delta \rho g \kappa_g}{\overline{\mu}} \left( \frac{\kappa_z^e}{\kappa_g} \right) \left( \frac{\kappa_x^e}{\kappa_z^e} \right)^{\eta}, \tag{3}$$

where  $\kappa_x^e$  is the equivalent intrinsic permeability along the horizontal direction, and the exponent  $\eta$  describes the impact of permeability anisotropy. The last term represents the anisotropic effect described by the horizontal to vertical equivalent permeability ratio. By conducting numerical simulations of GDC dissolution in homogeneous media with embedded horizontal barriers, which is similar to the aforementioned method used by Elenius and Gasda (2013), Green and Ennis-King (2014) found that  $\eta=0.5$ , which indicates that dissolution increases with horizontal to vertical equivalent permeability ratio. Note that when  $\eta=0.5$ , the dissolution rate is actually proportional to the geometric mean of the permeabilities  $\sqrt{\kappa_x^e \kappa_z^e}$ . In contrast, Erfani et al. (2022) give  $\eta=0.21$ . A summary of  $\eta$  is listed in Table 2. However, the results from Soltanian et al. (2017) and Elgahawy and Azaiez (2021) show that the increasing  $\kappa_x^e$  at fixed  $\kappa_z^e$  can reduce the asymptotic dissolution rate, implying that  $\eta$  may be negative.

## 4 Computational Approach

## 4.1 Model Setup

We conduct numerical simulations of Gravity-Driven Convection (GDC) over two-dimensional vertical fields of various permeability heterogeneity, which varies in space as a function of the horizontal and vertical distances. We assume that the interface between the  $CO_2$ -rich phase and brine is horizontal under buoyancy force, and that the brine at the interface is always saturated with  $CO_2$ . We only study the portion below the interface and therefore all simulations are conducted with a single-phase model. Initially, the  $CO_2$  concentration in brine is very low and the system is at static state.  $CO_2$  enters into the domain through the top boundary that has fixed high  $CO_2$  mass fraction. Brine is represented by a high-concentration Sodium Chloride (NaCl) solution. The objective is to get a quantitative relation between representative properties of the heterogeneous field and the  $CO_2$  dissolution rate through the top boundary.

Figure 2 shows the two-dimensional vertical domain used to simulate the development of dissolution process enhanced by GDC. Detailed parameter settings are as follows. The top boundary, which represents the interface between brine and  ${\rm CO}_2$ -rich phase, has constant liquid pressure  $p_0=150$  [bar] and constant  ${\rm CO}_2$  mass fraction  $X_0^C=0.041$  [kg·kg $^{-1}$ ]. The bottom and lateral boundaries are no-flow boundaries. Initially, the brine has a very low background  ${\rm CO}_2$  mass fraction  $X_{bg}^C\approx 0.0006$  [kg·kg $^{-1}$ ]. An initial brine pressure of  $p_0=150$  [bar] is imposed at the top layer and increases downwards at a gradient of  $\rho g$ , which means that the brine is initially at hydrostatic state. The temperature  $(T_c)$  is fixed to 60 [°C], and the salinity of brine  $(m^S)$  is constant 0.5 [molal]. When the mass fraction of aqueous  ${\rm CO}_2$  increases from initial  $X_{bg}^C$  to  $X_0^C$  on the top boundary, the density of brine increases by  $\Delta \rho = 8.2$  [kg·m $^{-3}$ ]. We note that the viscosity of the brine slightly changes from 1.0 [mpa·s] at the minimum background  ${\rm CO}_2$  mass fraction  $(X_0^C)$ . Thus, the mean viscosity is approximated by  $\overline{\mu} = 0.95$  [mpa·s]. Table 3 summarizes the parameters that are used in the numerical simulations.

We note that the top boundary condition is different from most of those used in literature where the convection of the CO<sub>2</sub>-saturated layer is either totally or partially suppressed (see Table 1). For instance, in Pau et al. (2010), the top constant concentration boundary only allows CO<sub>2</sub> to enter the domain via diffusion; this top boundary condition generates a much lower dissolution rate. In some studies the top boundary is partially permeable and larger dissolution rates are obtained (Hesse, 2008; Elenius et al., 2014; Rasmusson et al., 2015). Although it is more realistic to add a capillary transition zone beneath the top boundary (Elenius et al., 2015), experimental results with totally permeable top boundaries (Rasmusson et al., 2017) show only little discrepancy from the numerical results obtained in a model that includes the capillary transition zone (Martinez and Hesse, 2016). Therefore, we employ the single phase flow model with permeable top boundary in this study, although our model is capable of two-phase flow simulations (Wang et al., 2022).

An initial perturbation of the initial CO<sub>2</sub> mass fraction on the top boundary is added to stimulate the onset of instability at the beginning of simulation. We added a white noise that follows an uncorrelated Gaussian distribution (Fu et al., 2013). The magnitude of the noise is 1% of the initial mass fraction. Hidalgo and Carrera (2009) show that instability can be generated by the numerical error without introducing any external noise. Certainly, even though larger noise strength tends to accelerate the appearance of instability fingers, it is however unlikely to change the statistic behavior of the dissolution rate once the

**Figure 2.** Sketch of setup design. The size of the simulation domain is  $B \times L = 7.5 \times 7.5$  [m<sup>2</sup>], and other hydrogeology properties are summarized in Table 3.

**Table 3.** Summary of the parameters adopted during numerical simulations.

| Parameters                                            | Symbol           | Units                                  | Values                |
|-------------------------------------------------------|------------------|----------------------------------------|-----------------------|
| Domain size                                           | $L \times B$     | $[m \times m]$                         | $7.5 \times 7.5$      |
| Grid discretization                                   | $n_x \times n_z$ | [-]                                    | $100 \times 100$      |
| Porosity                                              | $\phi$           | [-]                                    | $0.15^{c,f}$          |
| Geometric mean permeability                           | $\kappa_g$       | $[m^2]$                                | $10^{-12\ b,i}$       |
| Initial liquid pressure at top layer                  | $p_0$            | [bar]                                  | 150 <sup>h</sup>      |
| Initial background CO <sub>2</sub> (aq) mass fraction | $X_{bg}^{C}$     | $[kg \cdot kg^{-1}]$                   | $\sim 0.0006^{g,j}$   |
| CO <sub>2</sub> (aq) mass fraction at top boundary    | $X_0^C$          | $[kg \cdot kg^{-1}]$                   | 0.041                 |
| Brine density at top boundary                         | $ ho_0$          | $[kg \cdot m^{-3}]$                    | 1017                  |
| Brine density increase due to increased               | $\Delta  ho$     | $[kg \cdot m^{-3}]$                    | 8.2                   |
| CO <sub>2</sub> (aq) mass fraction                    | $\Delta \rho$    | [kg·III ]                              | 0.2                   |
| Salinity                                              | $m^S$            | [molal]                                | $0.5^{~e,i}$          |
| Mean viscosity                                        | $\overline{\mu}$ | [mpa·s]                                | 0.95                  |
| Molecular diffusion coefficient                       | D                | $[\mathrm{m}^2 \cdot \mathrm{s}^{-1}]$ | $2\times10^{-9\ d,i}$ |
| Temperature                                           | $T_c$            | [°C]                                   | $60^{a,k}$            |

Reference: <sup>a</sup> Spycher et al. (2003); <sup>b</sup>Chadwick et al. (2004); <sup>c</sup> Maldal and Tappel (2004);

instability has fully developed (Hidalgo and Carrera, 2009; Elenius and Johannsen, 2012). We note that in a more realistic 3D scenario, the dissolution rate may be approximately 25% higher than that observed in 2D cases. However, this difference is relatively minor when compared to the significant variability in permeability commonly encountered in geologic media (Wang et al., 2022).

<sup>&</sup>lt;sup>d</sup>Tewes and Boury (2005); <sup>e</sup>Spycher and Pruess (2005); <sup>f</sup> Mathieson et al. (2009);

<sup>&</sup>lt;sup>9</sup>Xu et al. (2007);<sup>h</sup>Iding and Ringrose (2010);<sup>i</sup>Elenius and Johannsen (2012);

<sup>&</sup>lt;sup>j</sup> Saaltink et al. (2013); <sup>k</sup> Strandli and Benson (2013).

## 185 4.2 Governing Equations

On the basis of the mass balances of water and CO<sub>2</sub> species, the two governing transport equations are constructed as,

$$0 = \frac{\partial(\phi \rho X^H)}{\partial t} + \nabla \cdot (\rho X^H \mathbf{q}) - \nabla \cdot (\phi \mathbf{D} \rho \nabla X^H), \tag{4}$$

$$0 = \frac{\partial(\phi \rho X^C)}{\partial t} + \nabla \cdot (\rho X^C q) - \nabla \cdot (\phi D \rho \nabla X^C), \tag{5}$$

where  $\phi$  [-] is the porosity of the saline formation,  $\rho$  [kg·m<sup>-3</sup>] represents the density of brine, X [kg·kg<sup>-1</sup>] is the mass fraction, superscripts (H,C) represent the water and aqueous  $CO_2$  species, respectively, t [s] denotes the time,  $\mathbf{D} = D\mathbf{I}_d$  [m<sup>2</sup>·s<sup>-1</sup>] denotes the dispersion tensor, which is assumed locally constant. Local dispersion has little impact on the asymptotic dissolution rates, which is the objective of this work (Prasad and Simmons, 2003; Hidalgo and Carrera, 2009). The discharge rate (q) is given by Darcy's law

$$q = -\frac{\kappa}{\mu}(\nabla p - \rho g \nabla z),$$
 (6)

where  $\kappa$  [m<sup>2</sup>] is the intrinsic permeability,  $\mu$  [pa·s] is the viscosity, p [pa] is the liquid pressure, and z [m] is the depth. Besides, we have the following constraints:

$$X^{S} = 0.05844X^{H}m^{S},\tag{7}$$

and

$$X^H + X^C + X^S = 1,$$
 (8)

where  $m^S$  denotes the molality of salt. Here, we assume that the salt comprises only NaCl, and the molality of NaCl  $(m^S)$  is fixed. Define  $\omega = (1 + 0.05844 m^S)$  and then Equations (7) and (8) merge to

$$\omega X^H + X^C = 1. (9)$$

Under isotherm and isosalinity condition,  $\rho$  and  $\mu$  are only governed by liquid pressure and aqueous  $CO_2$  mass fraction (see Appendix A).

## 4.3 Dimensionless variables

In order to facilitate the analysis, results are presented using the following dimensionless variables, which are defined based on the works of Ennis-King and Paterson (2003) and Rasmusson et al. (2017),

$$X^{C*} = \frac{X^C}{X_0^C}, \qquad X^{H*} = \frac{X^H}{X_0^H},\tag{10}$$

and

$$t^* = \frac{t}{t_c}, \qquad x^* = \frac{x}{L_c}, \qquad z^* = \frac{z}{L_c},$$
 (11)

where  $X_0^C$  and  $X_0^H$  are, respectively, the maximum  $CO_2$  and water mass fractions, and  $t_c$  and  $L_c$  are the characteristic time and length scale of the gravity-driven convection problem defined as

$$t_c = \frac{(\overline{\mu}\phi)^2 D}{(\Delta \rho g \kappa_q)^2}, \qquad L_c = \frac{\overline{\mu}\phi D}{\Delta \rho g \kappa_q}.$$
 (12)

The characteristic time  $t_c$  has been found to be closely related to the onset time of gravity-driven convection, and the characteristic length  $L_c$  closely related to the earliest finger width. In our simulations, we found that the earliest finger width, denoted as  $\ell_c$ , can be approximated by  $\ell_c \approx 70L_c$ . The governing equations are expressed in dimensionless form in the Appendix B. Importantly, by expressing Darcy's law and the global dissolution rate in dimensionless form, we obtain that

$$q^* = \frac{q}{q_c}, \qquad q_c = \frac{\Delta \rho g \kappa_g}{\overline{\mu}},\tag{13}$$

$$F^* = \frac{F}{F_c}, \qquad F_c = X_0^C \rho_0 \frac{\Delta \rho g \kappa_g}{\overline{\mu}}. \tag{14}$$

The characteristic velocity  $v_c = q_c/\phi$  is closely related to the finger-tip velocity (see Elenius and Johannsen (2012)). We note that although the vertical length scale (i.e., domain thickness) and related dimensionless number (e.g., Rayleigh number) have been typically used to study gravity instability in the literature (Rasmusson et al., 2017, and references therein), herein we do not use it because the vertical domain size has little impact on the asymptotic enhanced dissolution rate driven by instability fingers (Elenius et al., 2015; Tsinober et al., 2022). This definition of the dimensionless length scale without using the domain thickness indicates that the instability is a random statistic behavior that does not change with the domain size provided that the domain is large enough to accommodate sufficient number of density fingers. In the supplementary information, we have shown that increasing the vertical domain size employed in this work does not systematically affect the asymptotic dissolution rate (Elenius and Johannsen, 2012). Simmons et al. (2001) give a detailed discussion of the limitation of using Rayleigh number based on the domain thickness.

## 4.4 Heterogeneity

The intrinsic permeability is the only aquifer property considered to vary in space. We represent the natural logarithm of the intrinsic permeability, denoted as  $Y = \ln \kappa$ , by a random space function to create multiple realizations of the aquifer

**Table 4.** Permeability heterogeneity adopted for the numerical simulations. The vertical correlation length is fixed to  $\lambda_z = 2L_c$ , and the horizontal correlation length is  $\lambda_x = \Omega \cdot \lambda_z$ .

| Case        | Ω        | $\sigma_Y^2$ |
|-------------|----------|--------------|
| Homogeneous | -        | -            |
| Isotropic   | 1        | 1            |
| Isotropic   | 1        | 4            |
| Anisotropic | 2        | 0.1          |
| Anisotropic | 2        | 1            |
| Anisotropic | 4        | 0.1          |
| Anisotropic | 4        | 1            |
| Anisotropic | 4        | 4            |
| Anisotropic | $\infty$ | 1            |

permeability distribution. The random space function model is characterized by an exponential covariance function with mean  $(\overline{Y})$ , variance  $(\sigma_Y^2)$ , horizontal correlation length  $(\lambda_h)$  and vertical correlation length  $(\lambda_v)$ . The geometric mean of the intrinsic permeability is fixed to  $\kappa_g = 10^{-12} \text{ m}^2$ . Different degrees of heterogeneity and correlation anisotropy  $\Omega = \lambda_h/\lambda_v$  are explored with  $\sigma_Y^2$  ranging between highly homogeneous,  $\sigma_Y^2 = 0.1$ , to relatively highly heterogeneous,  $\sigma_Y^2 = 4$ , and  $\Omega$  ranging between isotropic,  $\Omega = 1$ , and perfectly stratified,  $\Omega = \infty$ . The perfectly stratified random field is formed by separate horizontal layers of constant properties. The vertical correlation length is fixed to  $\lambda_z = 2L_c$ , and the horizontal correlation length is  $\lambda_x = \Omega \cdot \lambda_z$ . For comparison purposes, we also considered a homogeneous medium with  $\kappa = \kappa_g$ . In total, we conduct GDC simulations with 554 realizations. Random fields were generated using the sequential Gaussian simulation method implemented in the SGSIM code of GSLIB (Journel and Huijbregts, 1976; Deutsch et al., 1992). Table 4 summarizes the statistical properties of the random fields. An illustrative realization of each random field type is shown in Figure 4.

#### 4.5 Global Measures

Simulation results are analyzed based on two global measures of the dissolution behavior. The global dissolution rate (F [kg·m<sup>-2</sup>·s<sup>-1</sup>]) is defined as the rate at which  $CO_2$  dissolves from the upper boundary at z=0. This can be expressed as (Hidalgo and Carrera, 2009)

$$F(t) = \frac{1}{L} \int_{0}^{L} \left[ \rho X^{C} q_{z} - \phi D \rho \frac{\partial X^{C}}{\partial z} \right]_{z=0} dx.$$
 (15)

Initially, the domain is stable, the convection flux is zero, and only molecular diffusion transports downwards the dissolved CO<sub>2</sub>. The density-driven unstable convection does not emerge until the CO<sub>2</sub> mass fraction distribution reaches a critical point. After this, the dissolution rate rapidly increases to a quasi constant value until the domain is almost totally saturated with aqueous CO<sub>2</sub>. In our simulations, the asymptotic value of the global dissolution rate  $F_{\infty}$  is estimated as the temporal average

of F(t) over the period of  $[t_b/3, t_b]$ , where  $t_b$  is the time when the earliest finger of aqueous  $CO_2$  reaches the bottom (time when the maximum bottom  $CO_2$  mass fraction exceeds 25% of  $X_0^C$ ).

Another important global parameter that describes the vertical migration or the penetration depth of the  $CO_2$  plume is the vertical finger-tip velocity (Prasad and Simmons, 2003). The representative vertical finger-tip velocity of the  $CO_2$  plume is estimated as.

$$v(t) = \max_{0 < z < B} \left\{ \frac{1}{L} \int_{0}^{L} \frac{1}{\phi} | q_z | dx \right\}.$$
 (16)

Figure 3 illustrates the concept of vertical finger-tip velocity in our simulations. Similar to the global dissolution rate behavior, the vertical finger-tip velocity also reaches an asymptotic value (Elenius and Johannsen, 2012) in the constant-flux regime and remains at that value until the field is almost saturated. The asymptotic vertical finger-tip velocity  $v_{\infty}$  is also estimated by the temporal average of v(t) over the time interval  $[t_b/3, t_b]$ .

In order to characterize the overall hydraulic behavior of the permeability field, we estimated the equivalent permeability along the horizontal  $\kappa_x^e$  and vertical  $\kappa_z^e$  direction for each realization of the random fields. For this, to estimate  $\kappa_i^e$  (i=x,y), we neglect gravity and saturate the porous medium with only water. We then set the domain sides perpendicular to the ith direction as impermeable, and we impose a pressure decrement  $|\Delta_i p|$  along the ith direction.  $\kappa_i^e$  is estimated by the total volumetric flow  $Q_i$  passing through the system in the ith direction as  $\kappa_i^e = \mu Q_i L_i/(A_i |\Delta_i p|)$ , where  $L_i$  is the domain size along the ith direction and  $A_i$  the corresponding cross-sectional area (Knudby and Carrera, 2005; Wang, 2022).

## **4.6** Numerical Features

The model is implemented in a Matlab reservoir simulator toolbox designed for CO<sub>2</sub> storage (Wang, 2022). The program is based on the finite volume method. The two governing equations (4) and (5) are solved simultaneously with an implicit Newton-Raphson method. Two-point flux approximation with up-winding scheme is used to calculate mass transport. Although the unconditional stable implicit method is employed, we should still control the time step in the numerical simulation to avoid significant numerical dispersion. In this work, the time step is limited by either advection  $(\Delta t 

Figure 3. Maps of the dimensionless  $CO_2$  mass fraction  $(X^{C*})$  and its horizontal average  $(\overline{X^{C*}})$ , the dimensionless vertical flow velocity  $(q_z^*/\phi)$  and the horizontal average of absolute dimensionless vertical flow velocity  $(\overline{|q_z^*/\phi|})$ , and temporal the development of dimensionless dissolution rate  $(F^*)$  and finger velocity  $(v^*)$  in an illustrative realization. The dimensionless finger velocity is represented by  $v^* = \max |\overline{q_z^*/\phi}|$ .

## 4.7 Effective Asymptotic Dissolution Models

Two effective dissolution models are proposed here based on previous results reported in literature (see Section 3) with the objective to offer a simplified representation of the overall asymptotic dissolution behavior in naturally occurring heterogeneous porous media. In the first effective model, we have extended the formula (3) to a more general expression that incorporates the effect of permeability anisotropy. In dimensionless form, the model expresses that

$$F_{\infty}^* = \gamma_1 \left(\frac{\kappa_z^e}{\kappa_q}\right)^{\alpha_1} \left(\frac{\kappa_x^e}{\kappa_z^e}\right)^{\beta_1}.$$
 (17)

Here, the dimensionless asymptotic dissolution rate is  $F_{\infty}^* = F_{\infty}/F_c$ . The last term on the right hand side of this expression represents the anisotropy of the permeability field, defined as  $a_f = \kappa_x^e/\kappa_z^e$ . Existing effective asymptotic dissolution models rely exclusively on Equation (17).

Alternatively, we explore whether predictions of the dissolution rate can be made using the finger-tip velocity  $v_{\infty}$ . This velocity refers to the rate at which the fingers or plumes of dissolved  $CO_2$  move downwards through the subsurface, and it can be observed using optical fiber sensors. Due to their inherent advantages—robustness, high sensitivity, compact size, and low signal loss—in situ optical fiber sensors have found widespread application in GCS and other subsurface projects (Bao et al., 2013; Wang et al., 2016; Joe et al., 2018, 2020; Stork et al., 2020; Sun et al., 2021; Kim et al., 2022; Liu et al., 2024; Mondanos et al., 2024). In virtue of measured  $CO_2$  concentration, we can easily predict the finger-tip velocity (Bogue, 2011; Bao et al., 2013). For this, in the second model, we have considered the following relationship written in dimensionless form as

$$F_{\infty}^* = \gamma_2 \left(\frac{v_{\infty}}{v_c}\right)^{\alpha_2} \left(\frac{\kappa_x^e}{\kappa_z^e}\right)^{\beta_2}.$$
 (18)

Essentially, this expression replaces the equivalent vertical permeability with the vertical finger velocity, which seems to offer a more direct description of the  $CO_2$  plume migration.  $\gamma_1$ ,  $\gamma_2$ ,  $\alpha_1$ ,  $\alpha_2$ ,  $\beta_1$  and  $\beta_2$  are tuning coefficients of the two effective asymptotic dissolution models.

## 5 Results and Discussion

#### 5.1 Impact of Heterogeneity

We first provide a general description of the impact of heterogeneity on the vertical migration of the CO<sub>2</sub> plume and dissolution rates. We focused on the influence of anisotropy in the correlation structure of permeability Ω and the degree of heterogeneity σ<sub>Y</sub><sup>2</sup>. Figure 4 shows the temporal evolution of the spatial distribution of CO<sub>2</sub> mass fraction for different types of heterogeneity. For illustrative purposes, we chose a representative permeability realization for each case. These realizations are shown in panel (a) of Figure 4, from which we can see that CO<sub>2</sub> fingering is strongly affected by heterogeneity. In particular, the presence of vertical well-connected high permeability zones (preferential channels) facilitates the initiation and growth of the instability

fingers (see for instance the second column of Figure 4). Actually, the white randomness of the top  $CO_2$  mass fraction (needed in homogeneous media to create instabilities) is redundant in heterogeneous porous media as instabilities are controlled by these vertical preferential channels. In all cases, results show that instability makes  $CO_2$  fingers grow, merge and re-initiate as also observed in laboratory experiments (Rasmusson et al., 2017; Liyanage, 2018; Tsinober et al., 2022) and numerical simulations (Elenius et al., 2015). In accordance with Simmons et al. (2001), we also see that heterogeneity dissipates vertical finger growth through dispersive mixing. This effect increases with  $\Omega$ , i.e., when horizontal well-connected high-permeability structures exist. This is strongly manifested in perfectly stratified media with  $\sigma_V^2 = 1$  and  $\Omega = \infty$  (fourth column of Figure 4).

For completeness, we also show the temporal evolution of the dissolution rate as a function of  $\sigma_Y^2$  and  $\Omega$  in Figure 5. Results are presented in terms of the ensemble average and the coefficient of variation of  $F^*(t^*)$ . As expected, in homogeneous media, the dissolution process shows the three well-known stages: an initial diffusion-controlled decrease, followed by an onset of nonlinear growth at  $t^* = 500$  due to instability, and eventually stabilizing at an approximately constant dissolution rate. However, in heterogeneous media, results show in all cases that the early-time evolution of  $F^*(t^*)$  in heterogeneous media is remarkably different than that in homogeneous media. The diffusive and the flux-growth regimes cannot be distinguished anymore and the system seems to be controlled by the interaction between gravity-driven convection and heterogeneity, indicating that heterogeneity helps triggering the onset of instability. This is consistent with results reported by Schincariol et al. (1997) and Simmons et al. (2001). Of course, this also indicates that caution is needed when using the onset time of instability for homogeneous media (Ennis-King and Paterson, 2005; Pruess and Zhang, 2008) in real applications.

The influence of  $\sigma_Y^2$  and  $\Omega$  can also be seen from Figure 5. Interestingly, in statistically isotropic heterogeneous media, the degree of heterogeneity  $\sigma_Y^2$  significantly influences the early behavior of  $F^*(t^*)$ , eventually converging to a similar asymptotic dissolution rate at large times. This suggests that the asymptotic dissolution rate might be governed by the existence of well-connected high-permeability zones, regardless of the specific high value of permeability. This effect is not seen in anisotropic heterogeneous media where we found that the higher the  $\sigma_Y^2$  the lower is the asymptotic dissolution rate. We attribute this to the fact that, when  $\Omega > 1$ , an increase in  $\sigma_Y^2$  produces stronger well-connected permeability layers that inhibits gravity-driven convection. For the same reason, for equal  $\sigma_Y^2$ , the higher permeability anisotropy  $\Omega$  the less significant is the asymptotic dissolution rate. We also report in this figure a measure of the uncertainty in  $F^*(t^*)$  given by the coefficient of variation (CV). We can observe that the coefficient of variation reaches a similar asymptotic value for all cases, regardless of the degree of heterogeneity. A similar trend is also observed for the vertical finger velocity, as shown in the second column of Figure 5, indicating a close relation between the dissolution rate and the vertical finger velocity.

## 5.2 The Effective Asymptotic Dissolution Rate

320

325

330

335

The tuning coefficients of the two effective asymptotic dissolution models were independently adjusted by regression analysis of all simulation data obtained from the 554 realizations of the permeability distributions. These realizations involved random fields with different correlation structures of permeability and degrees of heterogeneity. To do this, we used Ordinary Least Squares (OLS) regression of the natural logarithm of the dissolution rate models (17) and (18). We respectively obtained a coefficient of determination  $R^2$  of 0.71 and 0.84. The significance of all regression coefficients was below 0.05, meaning

Figure 4. (a) Logarithm permeability  $(\log(\kappa))$  distribution. (b) Development of dimensionless  $CO_2$  mass fraction distribution  $(X_l^{C^*})$  due to gravity-driven convection.

Figure 5. The temporal evolution of the ensemble average of the dimensionless dissolution rate  $(\langle F^* \rangle)$  through the top boundary and ensemble average of the dimensionless finger velocity  $(\langle v^* \rangle)$  for all the test cases listed in Table 4. Here, we also give the coefficient of variation (CV), which is the ratio of standard division to the ensemble average.

**Table 5.** Coefficients for effective asymptotic dissolution models obtained from OLS regression of all simulation data (see Supplementary table 1); the results from literature are also listed for comparison.

| Predictors | Coefficients                    | Values                | MAE | RMSE | Source                      |
|------------|---------------------------------|-----------------------|-----|------|-----------------------------|
| (17)       | $(\gamma_1, \alpha_1, \beta_1)$ | $(0.08, 1.1, -0.2)^a$ | 27% | 22%  | This work                   |
| (18)       | $(\gamma_2, \alpha_2, \beta_2)$ | $(0.34, 1.0, -0.3)^b$ | 20% | 19%  | This work                   |
| (17)       | $(\gamma_1, \alpha_1, \beta_1)$ | $(0.09^c, 1.0, 0)$    | 40% | 24%  | Elenius and Gasda (2013)    |
| (17)       | $(\gamma_1, \alpha_1, \beta_1)$ | $(0.09^c, 1.0, 0.5)$  | 76% | 44%  | Green and Ennis-King (2014) |

<sup>&</sup>lt;sup>a</sup> 95% confidence intervals for these three values are [0.079,0.084], [1.0,1.2] and [-0.26,-0.14], respectively.

that both models can properly explain dissolution rates. The results indicate that employing an upscaled permeability field with equivalent permeability does not compromise the depiction of dissolution efficiency in GDC simulations, although permeability upscaling does alter the shapes of the dissolution profiles. Table 5 provides a summary of the regression analysis. Similar to the values reported in the literature, we find that  $\alpha_1 = 1.1$ , which is close to unity, meaning that the effective dissolution efficiency is almost linearly proportional to the equivalent vertical permeability. Moreover, we find that, in anisotropic heterogeneous media, the anisotropy of the equivalent permeability ( $a_f = \kappa_x^e/\kappa_z^e$ ) can reduce the effectiveness of dissolution with a power law behavior given by  $\beta_1 = -0.2$ . This contradicts previous results obtained in homogeneous media with embedded horizontal impermeable inclusions (Green and Ennis-King, 2014; Erfani et al., 2022), which indicated that  $\beta_1 > 0$ .

Figure 6 compares the asymptotic dissolution rate predicted by the proposed asymptotic dissolution rate models, expressions (17) and (18), with corresponding simulated values. For completeness, we also show the performance of the reported expressions given by Elenius and Gasda (2013) and Green and Ennis-King (2014). We visually differentiate between isotropic and anisotropic random fields. We can see that the predictor given by Green and Ennis-King (2014) significantly overestimates the dissolution rate in anisotropic random fields. Actually, the expression by Green and Ennis-King (2014) does not seem to improve the prediction given by Elenius and Gasda (2013). The second effective dissolution model given by equation (18), proposed here based on the vertical finger-tip velocity, shows the best performance, indicating that the dissolution rate has a strong relationship with the finger-tip velocity. The Mean Absolute Error (MAE) for formulas (17) and (18) are 27% and 20%, respectively. These error are well accepted considering that even in homogeneous fields, the dissolution rate can fluctuate around 15% (Pau et al., 2010).

In this work, we found a negative impact of the anisotropy of the permeability field  $a_f$  on dissolution rates. This can be physically explained in the following manner. Instabilities in the form of fingers exhibit nonuniform periodic high concentration distributions along the horizontal direction. When the spatial continuity of permeability values in the horizontal plane is substantial, any nonuniform concentration in this direction is rapidly eradicated by the enhancement of horizontal mixing induced by the introduction of companion horizontal flows. Consequently, the formation of finger-like instabilities becomes

<sup>&</sup>lt;sup>b</sup> 95% confidence intervals for these three values are [0.32,0.37], [0.98,1.09] and [-0.37,-0.28], respectively.

<sup>&</sup>lt;sup>c</sup> Here, we update the value for the reference dissolution rate in the homogeneous case, because the original value was around 0.02 based on the impermeable top boundary (cf. Table 1).

more challenging, especially in scenarios with high horizontal permeability values. In accordance, Schincariol et al. (1997) show that increasing the horizontal correlation length of the permeability distribution (increase in  $\kappa_x^e$ ), can effectively inhibit instability growth and stabilize perturbations. Through numerical simulations in a wide variety of heterogeneous fields, Simmons et al. (2001), Soltanian et al. (2017) and Elgahawy and Azaiez (2021) also conclude that instability is dampened when  $\kappa_x^e/\kappa_z^e$  is relatively large. Recent studies by Tsinober et al. (2022) and Hansen et al. (2023) have also highlighted that the introduction of a minor horizontal background flow in geological carbon sequestration fields enhances horizontal mixing. This enhancement of mixing disrupts the growth of fingers, consequently leading to a reduction in the dissolution rate. All these works also suggest that horizontal flows have the potential to decrease nonuniform instabilities. To better illustrate this, Figure 7 shows simulated and estimated effective dissolution rates as a function aquifer properties  $\{\kappa_z^e, \kappa_x^e/\kappa_z^e, \text{ and } v_\infty^*\}$ . The figure shows a clear negative dependence of the asymptotic dissolution rates with permeability anisotropy. From Figure 7, it is also evident that the performance of our predictors is not influenced by permeability. This suggests that the findings of this study can be extended to fields with greater permeability heterogeneity.

We acknowledge that our numerical simulations have not covered cases with  $a_f < 1$ , which are uncommon in natural sediment formations. Studies conducted by Simmons et al. (2001) have demonstrated that vertically stratified structures with  $a_f < 1$  can encourage vertical unstable convection by diminishing horizontal dissipation in instability fingers. Thus, it is observed that instability becomes more pronounced when  $a_f$  is small, aligning with the findings of our study.

## 6 Conclusions

GCS in saline aquifers reduces the release of CO<sub>2</sub> into the atmosphere, thereby mitigating its impact on climate change. Once CO<sub>2</sub> is injected, the less dense CO<sub>2</sub>-rich phase floats over the brine and gets trapped beneath an impermeable geological formation. At the interface between the brine phase and the CO<sub>2</sub>-rich phase, CO<sub>2</sub> slowly dissolves into the brine, thereby reducing the risk of CO<sub>2</sub> leakage. Estimating CO<sub>2</sub> dissolution rates in this context is complex, as it requires characterizing the enhancement of dissolution due to Gravity-Driven Convection (GDC), which creates instability fingers that transfer the high CO<sub>2</sub> concentration brine downwards. While many studies offer a deep understanding of GDC in homogeneous porous media, less is known about dissolution rates in more realistic heterogeneous porous media. In this work, we have systematically analyzed the effect of heterogeneity on GDC-driven dissolution rates during GCS. To achieve this, we have conducted numerical simulations of GDC in multiple aquifer realizations of various permeability distributions. These distributions follow a random space function that exhibits distinct correlation structures (anisotropy) and degrees of heterogeneity.

Based on these simulations, we have explored the impact of heterogeneity on the temporal evolution of dissolution rates. We find that in heterogeneous porous media, vertical preferential channels, formed by well-connected high-permeability zones, play a significant role in initiating and developing instability fingers. Moreover, the presence of horizontal well-connected high-permeability structures inhibits the vertical growth of fingers by favoring dispersive mixing. Consequently, in anisotropic heterogeneous porous media, an increase in the degree of heterogeneity leads to a decrease in the asymptotic dissolution rate.

**Figure 6.** Comparison of the performances of the predictors given by Elenius and Gasda (2013), Green and Ennis-King (2014) and this work (cf. Table 5).

**Figure 7.** Simulated and estimated effective dissolution rates as a function aquifer properties.

We have developed two effective asymptotic dissolution rate models derived from regression analysis of all the simulated data. The first model estimates the asymptotic dissolution rate using the aquifer's general hydraulic properties. For this model, we achieved a coefficient of determination ( $R^2$ ) of 0.71, indicating a strong correlation between the variables compared to previous effective dissolution models (Elenius and Gasda, 2013; Green and Ennis-King, 2014),

$$F_{\infty} = 0.08 X_0^C \rho_0 \frac{\Delta \rho g \kappa_g}{\overline{\mu}} \left(\frac{\kappa_z^e}{\kappa_g}\right)^{1.1} \left(\frac{\kappa_x^e}{\kappa_z^e}\right)^{-0.2}.$$
 (19)

This model requires some knowledge of the equivalent permeability value along the x and z directions  $\{\kappa_x^e, \kappa_z^e\}$  and fluid properties. The equivalent permeability can be estimated by a wide variety of methods, including hydraulic tests (see Renard and de Marsily (1997); Sanchez-Vila et al. (2006) for a review).

Alternatively, asymptotic dissolution rates can be estimated by the  $CO_2$  finger-tip velocity v. Results have demonstrated that the finger-tip velocity offers a better estimate of dissolution rates with a coefficient of determination ( $R^2$ ) of 0.84,

$$F_{\infty} = 0.34 X_0^C \rho_0 \frac{\Delta \rho g \kappa_g}{\overline{\mu}} \left( \frac{v_{\infty}}{v_c} \right)^{1.0} \left( \frac{\kappa_x^e}{\kappa_z^e} \right)^{-0.3}. \tag{20}$$

These effective asymptotic dissolution rate models express that the anisotropy of the permeability field (last term of the expressions) negatively affects dissolution rates. When permeability values have substantial spatial continuity horizontally, the corresponding increase in horizontal mixing inhibits nonuniform concentrations, making it harder for instabilities to form. These results differ from those presented by Green and Ennis-King (2014) in homogeneous media with horizontal barriers, where the dissolution rate is proposed to be enhanced by permeability anisotropy with an exponent of 0.5. In accordance with our work, Soltanian et al. (2017) and Elgahawy and Azaiez (2021) demonstrated that increasing  $\kappa_x^e$  at fixed  $\kappa_z^e$  can reduce the asymptotic dissolution rate.

The results from this study may have potential application to other common gravity-driven convection problems, such as contaminant migration, geothermal exploitation, saltwater intrusion and mineral precipitation/dissolution, where density differences may exist (Zhang and Schwartz, 1995; Simmons et al., 1999; Nield et al., 2008).

Supporting information. The supporting information can be accessed from https://zenodo.org/records/14061632

Code and data availability. The code and data are open source and can be accessed via Zenodo at the following URLs: https://zenodo.org/records/5833962 and https://zenodo.org/records/14061632.

Author contributions. Y. W.: Modelling, coding and writing; D. F. G.: Modelling, writing and supervising; M. W. S.:Modelling, writing and supervising.

Competing interests. The authors have no competing interests to declare that are relevant to the content of this article

Acknowledgements. We acknowledge the help from Oriol Bertran-Oller and Rodrigo Perez in accessing the TITANI- high performance green computing cluster of the civil engineering school. This work was partially supported by the European Commission, through project MARSOLUT (grant H2020-MSCA-ITN-2018); by the Spanish Ministry of Economy and Competitiveness, through project MONOPOLIOS (RTI 2018-101990-B-100, MINECO/ FEDER); and by the Catalan Agency for Management of University and Research Grants through FI 2017 (EMC/2199/2017).

# Appendix A: Density and Viscosity

In the numerical model, instead of using a simplified linear expression of the brine density based on the aqueous CO<sub>2</sub> concentration (e.g., Elenius et al., 2015; Martinez and Hesse, 2016), we incorporated a more realistic brine density expression derived from (Vilarrasa, 2012), given the sensitivity of instability to the fluid property (Jafari Raad et al., 2016; Rasmusson et al., 2015, 2017). In this model, the brine density, *ρ* [kg· m<sup>-3</sup>], depends on brine phase pressure, temperature, molality of NaCl and CO<sub>2</sub> concentration. The expression for brine is given by Garcia (2003)

$$\rho = \rho_{lr} + cM^C - c\rho_{lr}V_{\phi},\tag{A1}$$

where  $c \,[\text{mol·m}^{-3}]$  is the number of moles of  $CO_2$  per unit volume of brine phase;  $M^C \,[\text{kg·mol}^{-1}]$  is the molar mass of  $CO_2$ ; and  $V_{\phi} \,[\text{m}^3 \cdot \text{mol}^{-1}]$  is the apparent molar volume of  $CO_2$  given by

$$V_{\phi} = 3.751 \times 10^{-5} - 9.585 \times 10^{-8} T_c + 8.74 \times 10^{-10} T_c^2 - 5.044 \times 10^{-13} T_c^3, \tag{A2}$$

where  $T_c$  [°C] is temperature in Celsius;  $\rho_{lr}$  is the brine density when there is no  $CO_2$  dissolution, calculated by Phillips et al. (1982)

$$\rho_{lr} = -3.033405 \times 10^3 + 1.0128163 \times 10^4 \iota - 8.750567 \times 10^3 \iota^2 + 2.66310 \times 10^3 \iota^3, \tag{A3}$$

with

$$\iota = -9.9595 \exp(-4.539 \times 10^{-3} m^S) + 7.0845 \exp(-1.638 \times 10^{-4} T_c)$$

$$+3.9093 \exp(2.551 \times 10^{-10} p), \tag{A4}$$

where  $m^S$  [molal] is the molality of NaCl and p [pa] is the pressure of brine. Equation (A1) applies to  $5 < T_c < 297$  [°C] and  $p_{sv} [bar]. Equation (A3) applies to <math>10 < T_c < 350$  °C,  $0.25 < m_l^S < 5$  [molal] and  $p_{sv} [bar] (Phillips et al., 1982). Here, <math>p_{sv}$  is saturated vapor pressure. Rearranging Equation (A1), we have (Vilarrasa, 2012)

$$\rho = \rho_{lr} \frac{1}{1 - X^C f_{\delta}} \approx \rho_{lr} (1 + X^C f_{\delta}), \tag{A5}$$

with

$$f_{\delta} = 1 - \rho_{lr} \frac{V_{\phi}}{M^C}; \tag{A6}$$

here  $X^C$  denotes the mass fraction of  $CO_2$ . The viscosity of brine is calculated following the works of Garcia (2003) and Kumagai and Yokoyama (1999)

$$\mu = (3.85971 - 1.32561 \times 10^{-2} T_k) m^S + (-5.37539 + 1.90621 \times 10^{-2} T_k) (m^S)^{1/2}$$

$$+ (8.79552 - 3.17229 \times 10^{-2} T_k) m^C + (-7.22796 + 2.64498 \times 10^{-2} T_k) (m^C)^2$$

$$+ 1.69956 \times 10^{-9} (p - 1 \times 10^5) + \mu_w (T_k, p = 10^5 [Pa]),$$
(A7)

where  $T_k$  [K] is temperature in Kelvin,  $m^C$  [molal] is the molality of CO<sub>2</sub>, and  $\mu_w$  [mPa·s] is the viscosity of pure water.

## 460 Appendix B: Dimensionless Governing Equations

Given the dimensionless variables defined in section 4.2, the governing mass balance equations (4) and (5) can be written in dimensionless form as

$$\frac{\partial X^{C*}}{\partial t^*} = -\nabla^* \cdot \left( X^{C*} \boldsymbol{q}^* \right) + \nabla^* \cdot \left( \nabla^* X^{C*} \right), \tag{B1}$$

$$\frac{\partial X^{H*}}{\partial t^*} = -\nabla^* \cdot \left( X^{H*} \boldsymbol{q}^* \right) + \nabla^* \cdot \left( \nabla^* X^{H*} \right), \tag{B2}$$

where Darcy's law is expressed as

$$q^* = -\exp(Y')(\nabla^* p^* - \rho^* \nabla^* z^*),$$
 (B3)

and  $\nabla^* = [\partial/\partial x^*, \partial/\partial z^*]$ . Y' is the deviation of the natural log of the intrinsic permeability from the mean, i.e.,  $Y' = Y - \langle Y \rangle$ . The geometric mean permeability is  $\kappa_g = \exp(\langle Y \rangle)$ . The fluid pressure and the density are normalized by

$$p^* = \frac{p\kappa_g}{\mu\phi D}$$
, and  $\rho^* = \frac{\rho}{\Delta\rho}$ . (B4)

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
