# Peer review of "Numerical Analysis of the Effect of Heterogeneity on CO2 Dissolution Enhanced by Gravity Driven Convection"

_EGUsphere, 2024_

## Referee Comment (RC1)

**Review of paper "Numerical Analysis of the Effect of Heterogeneity on CO2 Dissolution Enhanced by Gravity Driven Convection" submitted to HESS**

General comments:

It is a pleasure procedure to review this interesting manuscript, which deals with the dissolution trapping of CO2 during geological carbon sequestration, an important means to reduce carbon emissions. In the process of sequestering CO2 in the deep saline aquifer, a concern is that the CO2 is less dense than the saline water, which gives rise to the possibility of upward CO2 leak. It is noticed that dissolution of CO2 in the brine is a crucial mechanism that reduces the risk of leak by transforming the supercritical CO2 into aqueous CO2. The significant challenge lies in the fact that the dissolution process is susceptible to instabilities driven by gravity and permeability heterogeneity, both of which are ubiquitous and involve big uncertainties. Therefore, it is important to characterize the dissolution rate of CO2 in the saline water, which is the objective of this manuscript.

Through reading the manuscript, I can feel that the authors have spent a big effort in preparing this work. They carefully selected the parameters for their numerical model based on the data from a deep literature review. They have conducted a large quantity of numerical simulations to build a data pool for systematic analysis. Finally, they give fruitful analysis and discussions of the results. Based on my own reading of the manuscript, I give my support of publication of this work on HESS. Several revisions are needed before it is accepted for publication.

Major comments:

(1)This manuscript improves the current predictor for enhanced CO2 dissolution due to gravity driven convection. Most of current predictors for enhanced CO2 dissolution address the homogeneous cases, which may have limitations for the real heterogeneous problem. Two representative works, as shown in Table 5 and Figure 6, have offered the preliminary predictor for the CO2 dissolution in heterogeneous fields. However, while one of them simplifies the predictor by neglecting the anisotropic effect, the other may not properly incorporate the ansotropic effect. The authors provided a new explanation of the anisotropic effect on the density driven dissolution. Their numerical results show that for a fixed vertical equivalent permeability increasing the horizontal equivalent permeability can reduce the dissolution efficiency, because increased horizontal permeability can increase horizontal mass exchange, make the horizontal mass distribution more uniform, and thus reduce the instability. My question is: I realize that the gamma_1=0.08 obtained by the data regression is quite similar to that for the 0.09 listed in Table 1. Do they have any relations? This is based on my observation that the alpha_1=1.1 is very close to 1.0 for the homogeneous.

(2)Furthermore, the authors introduced a new predictor using finger velocity, which is particularly intriguing because finger velocity can be measured using optical fiber technology. Given the rapid advancements and expanding applications of optical fibers in the field of geosciences, this novel formula could serve as a valuable tool for monitoring the trapping of CO2 through dissolution. My

question is: The regression value for gamma_2=0.34 is quite different from 0.08 or 0.09, could you please explain why the gamma_2 is so different from gamma_1? I am wondering if it is simply a result of the data regression or maybe it have physical explanations.

(3)In the study of density driven instability, the fingers are usually irregular, as shown in the Figure 4 of your manuscript. Could you please explain why the fingers in Figure 3 are quite uniform?

(4)I can understand that the authors and many other researchers use two-dimensional model in both laboratory and numerical simulation, because of the high cost of three-dimensional model. I do agree that the two-dimensional study is very useful, but it would be nice if the authors can provide some discussions on the three-dimensional effect on GDC. I know it is hard to design a new simulation of new parameters, so I would appreciate it if the authors could find some related three-dimensional studies and give a short comparison of the difference of three dimensional simulations and two dimensional simulations. This may give more confidence for readers using the results from this work.

Minor comments:

1-The supplementary materials provided important and comprehensive information about the numerical modeling, but it is not fully referred in the paper. Please, give more clear reference of the supplementary materials in the paper.

2-Line 195 and 205, the appendix is missing.

3-Line 294, it would be better if we say in panel (a) of Figure 4 rather than in the first panel of Figure 4.

4-Line 299, I am wondering if you added white randomness ('white noise' maybe) in the heterogeneous fields?

5-Line 306, please remove the comma in 'dissolution process, shows'.

6-Line 388, please remove the period before the references.

7-The overall structure of this manuscript is very nice, dividing the whole article into logical sections of proper titles. However, it may be better if the authors can reorganize the section '6 Conclusions'. We can see that in the section '5 Results and Discussion'. The authors first describe the general impact of heterogeneity on the development of instability, and then perform log-linear regressions of the simulation results. However, in the conclusion section the authors do not organize these results in the same order. Moreover, it would be better if the first paragraph is split so that we have a short summary of this work before writing the conclusions. This does not affect the comprehending of this work, but it would be nicer if the authors can reorganize the conclusions.

Ming Yang,
Tsinghua University

---

## Author Comment (AC1)

**Response to reviewer 1 (Dr. Yang)**

**General Comments**

*It is a pleasure procedure to review this interesting manuscript, which deals with the dissolution trapping of $CO_2$ during geological carbon sequestration, an important means to reduce carbon emissions. In the process of sequestering $CO_2$ in the deep saline aquifer, a concern is that the $CO_2$ is less dense than the saline water, which gives rise to the possibility of upward $CO_2$ leak. It is noticed that dissolution of $CO_2$ in the brine is a crucial mechanism that reduces the risk of leak by transforming the supercritical $CO_2$ into aqueous $CO_2$. The significant challenge lies in the fact that the dissolution process is susceptible to instabilities driven by gravity and permeability heterogeneity, both of which are ubiquitous and involve big uncertainties. Therefore, it is important to characterize the dissolution rate of $CO_2$ in the saline water, which is the objective of this manuscript.*

*Through reading the manuscript, I can feel that the authors have spent a big effort in preparing this work. They carefully selected the parameters for their numerical model based on the data from a deep literature review. They have conducted a large quantity of numerical simulations to build a data pool for systematic analysis. Finally, they give fruitful analysis and discussions of the results. Based on my own reading of the manuscript, I give my support of publication of this work on HESS. Several revisions are needed before it is accepted for publication.*

R: We are grateful for your recognition of the importance of our work and your supportive comments regarding the effort we have invested in this study, including the careful selection of parameters based on a deep literature review, the extensive numerical simulations conducted, and the detailed analysis and discussion of the results. We sincerely thank you for your time and constructive feedback, which has significantly improved the quality of our manuscript.

Below, we provide detailed responses to each comment, along with a clear explanation of the revisions made to the manuscript. All changes will be incorporated into the subsequent revised version.

**Major Comments**

(1) *This manuscript improves the current predictor for enhanced $CO_2$ dissolution due to gravity driven convection. Most of current predictors for enhanced $CO_2$ dissolution address the homogeneous cases, which may have limitations for the real heterogeneous problem. Two representative works, as shown in Table 5 and Figure 6, have offered the preliminary predictor for the $CO_2$ dissolution in heterogeneous fields. However, while one of them*

*simplifies the predictor by neglecting the anisotropic effect, the other may not properly incorporate the ansotropic effect. The authors provided a new explanation of the anisotropic effect on the density driven dissolution. Their numerical results show that for a fixed vertical equivalent permeability increasing the horizontal equivalent permeability can reduce the dissolution efficiency, because increased horizontal permeability can increase horizontal mass exchange, make the horizontal mass distribution more uniform, and thus reduce the instability. My question is: I realize that the gamma_1=0.08 obtained by the data regression is quite similar to that for the 0.09 listed in Table 1. Do they have any relations? This is based on my observation that the alpha_1=1.1 is very close to 1.0 for the homogeneous.*

R: Your observation regarding the similarity between $\gamma_1 = 0.08$ and the value $\gamma = 0.09$ in Table 1 is insightful and highlights an important aspect of our work. The numerical proximity of $\gamma_1 = 0.08$ to $\gamma = 0.09$ arises because both parameters are rooted in the same underlying physics of gravity-driven convection. However, $\gamma_1 = 0.08$ is not a direct extension of the homogeneous value but rather a new parameter that incorporates the effects of heterogeneity. The similarity in $\gamma_1 = 0.08$ to the homogeneous value $\gamma = 0.09$ reflects the consistency between the homogeneous model and the heterogenous model. Interestingly, as mentioned by you, the data regression result shows that $\alpha_1 = 1.1$, which is very close to 1.0, again indicating the same underlying physics governing the vertical mass exchange driven by gravity-driven convection, i.e., the vertical mass flux due to density instability can be approximated by Darcian flux with permeability being represented by equivalent vertical permeability.

To clarify the relationship between these two values, we provide the following detailed explanation:

(i) $\gamma = 0.09$ in Table 1 is the value for the homogeneous field obtained by fitting the mean dissolution rate of 15 homogeneous cases using Equation (1) in the manuscript

$$F_\infty = \gamma X_0^C \rho_0 \frac{\Delta \rho g \kappa}{\bar{\mu}}. \tag{1}$$

(ii) $\gamma_1 = 0.08$ is obtained by fitting the dissolution rate for the heterogeneous fields using Equation (17) in the manuscript, which accounts for the effects of permeability anisotropy.

$$F_\infty^* = \gamma_1 \left(\frac{\kappa_z^e}{\kappa_g}\right)^{\alpha_1} \left(\frac{\kappa_x^e}{\kappa_z^e}\right)^{\beta_1}. \tag{17}$$

Considering $F_\infty^* = \dfrac{F_\infty}{X_0^C \rho_0 \frac{\Delta \rho g \kappa}{\bar{\mu}}}$, Equation (17) is equivalent to

$$F_\infty = \gamma_1 X_0^C \rho_0 \frac{\Delta \rho g \kappa}{\bar{\mu}} \left(\frac{\kappa_z^e}{\kappa_g}\right)^{\alpha_1} \left(\frac{\kappa_x^e}{\kappa_z^e}\right)^{\beta_1}. \tag{R1}$$

If the field is isotropically homogenous, i.e., $\kappa_z^e = \kappa_x^e = \kappa_g$, Equation (R1) should be

equivalent to Equation (1), which means $\gamma_1$ should be equal to $\gamma$ in theory. Our value $\gamma_1 = 0.08$ is very close to $\gamma = 0.09$, indicating the consistency of our theory and good predictivity of the formula.

We note that $\gamma$ value in our work is similar to those obtained in literature using open top boundaries (e.g., 0.075 in Elenius et al. (2012) using CTZ boundary, 0.065 in Martinez and Hesse (2016) using CTZ boundary, and 0.06 in Rasmusson et al. (2017) using permeable boundary). This similarity to some extent validates the reasonability of our numerical modeling. We also note the $\gamma$ value in our work is quite different from those research results obtained based on diffuse-only boundary. In our case, the convection of $CO_2$-saturated brine is allowed to pass the top boundary, which seems more realistic.

(2) *Furthermore, the authors introduced a new predictor using finger velocity, which is particularly intriguing because finger velocity can be measured using optical fiber technology. Given the rapid advancements and expanding applications of optical fibers in the field of geosciences, this novel formula could serve as a valuable tool for monitoring the trapping of $CO_2$ through dissolution. My question is: The regression value for gamma_2=0.34 is quite different from 0.08 or 0.09, could you please explain why the gamma_2 is so different from gamma_1? I am wondering if it is simply a result of the data regression or maybe it have physical explanations.*

R: The reviewer raises an excellent question regarding the significant difference between $\gamma_1 = 0.08$ and $\gamma_2 = 0.34$. This difference arises because $\gamma_1$ and $\gamma_2$ scale different physical quantities, as explained below:

$\gamma_1$ is obtained based on equivalent vertical permeability $\kappa_z^e / \kappa_g$, while $\gamma_2$ is based on dimensionless fingertip velocity ($v_\infty / v_c$). When the vertical permeability $\kappa_z^e / \kappa_g = 1$, the value of $v^* = v_\infty / v_c$ is around 0.3, as shown in bottom panel of Figure 3. This means there should be a factor of around 0.3 between $\gamma_1$ and $\gamma_2$. Interestingly, the value $\gamma_1 / \gamma_2 = 0.24$ is very close to the dimensionless fingertip velocity. This consistency makes the predictor physically sound.

(3) *In the study of density driven instability, the fingers are usually irregular, as shown in the Figure 4 of your manuscript. Could you please explain why the fingers in Figure 3 are quite uniform?*

R: Your observation about the uniformity of fingers in Figure 3 versus the irregularity in Figure 4 is meticulous. It is very common to see the irregular instability fingers in literature (e.g., Elenius & Gasda [2021] Farajzadeh et al. [2013]), which is also shown in our work. However, in the very early stage when the instability is just fully developed ( $t^* = 1500$ in Figure 3), the finger is quite uniform. These uniform fingers become irregular with time, as shown in Figure R1.

[Figure]

Figure R1: development of instability fingers (c.f., page 117 in Wang [2022], https://upcommons.upc.edu/handle/2117/376077).

(4) *I can understand that the authors and many other researchers use two-dimensional model in both laboratory and numerical simulation, because of the high cost of three-dimensional model. I do agree that the two-dimensional study is very useful, but it would be nice if the authors can provide some discussions on the three-dimensional effect on GDC. I know it is hard to design a new simulation of new parameters, so I would appreciate it if the authors could find some related three-dimensional studies and give a short comparison of the difference of three dimensional simulations and two dimensional simulations. This may give more confidence for readers using the results from this work.*

R: We agree with the reviewer that 3D effects are critical for field-scale applications of gravity-driven convection (GDC). While our study focuses on 2D simulations for computational efficiency, we recognize the importance of discussing 3D effects to provide a

more comprehensive understanding of the problem. We study recent researches that have successfully applied 2D-to-3D scaling relationships to bridge the gap between simplified models and real-world applications. The dissolution rate in a real 3D case will be higher than in the 2D cases employed in this work. As pointed out by Pau et al. [2010], the stabilized mass flux in the 3D scenario is observed to be 25% higher than that in a comparable 2D simulation. Although this difference is statistically significant, it is relatively minor considering the substantial variability in permeability often seen in geologic media. Therefore, the impact of additional spatial dimensions on both the onset time and the stabilized mass flux appears to be limited. In the revised manuscript, we will add a concise discussion on the differences between 2D and 3D simulations of GDC.

The following sentences will be added to the conclusion Section 4.1 . "We note that in the real 3D scenario, the dissolution rate may be approximately 25% higher than that observed in 2D cases. However, this difference is relatively minor when compared to the significant variability in permeability commonly encountered in geologic media Wang et al. (2022)." The revisions will be incorporated into the subsequent version.

**Minor Comments**

(1). *The supplementary materials provided important and comprehensive information about the numerical modeling, but it is not fully referred in the paper. Please, give more clear reference of the supplementary materials in the paper.*

R: We have added explicit references to the Supplementary Materials in Sections 2 and 4.3 to ensure readers are aware of the additional information available. The revisions will be incorporated into the subsequent version.

(2). *Line 195 and 205, the appendix is missing.*

R: Thank you. The appendix has been restored in the end of the manuscript. The revisions will be incorporated into the subsequent version.

(3). *-Line 294, it would be better if we say in panel (a) of Figure 4 rather than in the first panel of Figure 4.*

R: Thank you. We have revised to "panel (a) of Figure 4" for clarity. The revisions will be incorporated into the subsequent version.

(4). -Line 299, I am wondering if you added white randomness ('white noise' maybe) in the heterogeneous fields?

R: Yes. We have added a very small white noise in all simulations to trigger instability. However, we find this white noise has negligible effect in the instability development in the heterogeneous media, where the finger development is strongly influenced by the

permeability distribution.

(5). -Line 306, please remove the comma in 'dissolution process, shows'.

R: Thank you: We have removed the comma in "dissolution process shows". The revisions will be incorporated into the subsequent version.

(6). -Line 388, please remove the period before the references.

R: Thank you. We have corrected this formatting issue. Now we have moved the related sentence to section 4.7 to make the structure better. The revisions will be incorporated into the subsequent version.

(7). *-The overall structure of this manuscript is very nice, dividing the whole article into logical sections of proper titles. However, it may be better if the authors can reorganize the section '6 Conclusions'. We can see that in the section '5 Results and Discussion'. The authors first describe the general impact of heterogeneity on the development of instability, and then perform log-linear regressions of the simulation results. However, in the conclusion section the authors do not organize these results in the same order. Moreover, it would be better if the first paragraph is split so that we have a short summary of this work before writing the conclusions. This does not affect the comprehending of this work, but it would be nicer if the authors can reorganize the conclusions.*

R: Thank you. We have split the first paragraph into a brief summary and restructured the conclusions to mirror the Results section's logic. The revised conclusion now begins with a concise summary of the study's objectives and key findings, followed by detailed conclusions organized in the same order as the "Results" section. The revisions will be incorporated into the subsequent version.

Finally, we hope that these responses and the corresponding revisions have effectively addressed your concerns. We have also incorporated additional clarifications and refined the formatting throughout the manuscript. Once again, we extend our sincere thanks for your valuable feedback, which has played a crucial role in enhancing the quality of our work.

**Reference:**

Elenius, M. and Gasda, S. E.: Convective mixing driven by non-monotonic density, Transport in Porous Media, 138, 133–155, https://doi.org/10.1007/s11242-021-01593-3, 2021.

Farajzadeh, R., Meulenbroek, B., Daniel, D., Riaz, A., and Bruining, J.: An empirical theory for gravitationally unstable flow in porous media, Computational Geosciences, https://doi.org/10.1007/s10596-012-9336-9, 2013.

Pau, G. S., Bell, J. B., Pruess, K., Almgren, A. S., Lijewski, M. J., and Zhang, K.: High-resolution simulation and characterization of density-driven flow in CO2 storage in

saline aquifers, Advances in Water Resources, 33, 443 – 455, https://doi.org/https://doi.org/10.1016/j.advwatres.2010.01.009, 2010

Wang, Y.: Numerical Modeling of Geological Carbon Sequestration: Enhanced Dissolution in Randomly Heterogeneous Media, https://doi.org/10.5281/zenodo.6769788 , 2022 .

---

## Author Comment (AC2)

**Response to Reviewer 2:**

We would like to thank the reviewer for the positive feedback and thoughtful suggestions. Below are our responses to the comments raised:

**General comments**

*The article addresses the dissolution trapping of $CO_2$ during geological carbon sequestration in deep saline aquifers, a critical process that serves as a key indicator for evaluating the safety of long-term $CO_2$ storage. By trapping $CO_2$ in the aqueous phase, dissolution trapping helps prevent $CO_2$ leakage. However, the dissolution process can result in gravity-driven instabilities, such as Gravity-Driven Convection (GDC), and is influenced by formation properties like permeability and porosity, which are subject to uncertainties due to heterogeneities. The article aims to quantitatively analyze the explicit relationships between GDC-driven dissolution rates and permeability heterogeneities reported in the literature through numerical experiments.*

*The article is well-written and easy to understand. It is evident that the authors have carefully guided readers step by step through their work, supported by an in-depth literature review. Additionally, they have conducted a substantial number of numerical simulations across various setups to provide robust evidence. Based on my assessment of the manuscript, I believe this work is highly deserving of publication, subject to revisions.*

R: We are gratified that the reviewer has acknowledged the manuscript as well-written and clear. We sincerely appreciate the recognition of the meticulous explanations and the robustness of the numerical simulations conducted in this study. Additionally, we are thankful for your suggestion that the work merits publication, pending revisions. We are eager to address the comments to further enhance the clarity and impact of the manuscript. The following revisions will be incorporated into the subsequent version.

**Major Comments:**

*These are primarily comments from me and do not involve major revisions to the article.*

(1). *The 2D numerical setup is well-chosen, in my opinion, and all choices are clearly explained. Simulation data on a 100x100 grid over an extended period are obtained from 554 realizations of the permeability distributions. Additionally, the time-step size is governed by CFL conditions, which can result in a significant number of time steps. I assume this process is computationally intensive, and I was wondering how challenging it would be to extend this work to 3D, as doing so could provide valuable insights. Perhaps providing an approximate*

*value of the CPU time per simulation could serve as a good starting point and indicator.*

R: Thank you for raising the important point about extending our study to a 3D framework and acknowledging the computational challenges we have encountered. We fully agree that a 3D setup would offer a more realistic depiction of permeability heterogeneities in geological formations. However, the current computational resources pose significant limitations on the feasibility of conducting ensemble-based studies in 3D. As you correctly pointed out, the 2D simulations were already computationally intensive due to the large number of realizations (554) and the constraints imposed by the CFL time-step conditions. Extending these simulations to 3D would introduce considerable additional computational complexity.

To elaborate, the current 2D simulation, of which the grid discretization is 100×100, involves a $(2 \cdot 10^4) \times (2 \cdot 10^4)$ sparse matrix (here, 2 means we have two independent variables), taking approximately **4–7 hours** to complete. In contrast, a 3D simulation, of which the grid discretization is 100×100×100, would correspond to a $(2 \cdot 10^6) \times (2 \cdot 10^6)$ sparse matrix. Based on our estimates, the simulation time for a single 3D case would be at least 4×100 hours (around **17 days**). **Moreover, increasing the matrix size would also significantly elevate the memory requirements. Actually, we tried to give a 3-D example for you on my own desktop (not a cluster), but the simulation was extremely slow and it broke down with "Out of memory." warning, during estimating the condition number with LU using** `condest`**. The CPU and the RAM of the computer used are "12th Gen Intel(R) Core(TM) i9-12900K 3.20 GHz" and "64.0GB", respectively.** Given that we need to perform a total of 554 simulations, the computational burden and memory demands associated with a 3D setup would be formidable and currently beyond our capacity.

Second, it is worth noting that the stabilized mass flux in the 3D scenario is approximately 25% higher than that observed in a comparable 2D simulation (Pau, 2010). Although this difference is statistically significant, it is relatively minor when compared to the several orders of magnitude variability in permeability commonly seen in geological media (e.g., $\kappa = 10^{-16} - 10^{-12}$ m$^2$ and $\sigma^2_{\ln \kappa}$ = 3-10, as reported by Wang et al., 2022). This indicates that the impact of additional spatial dimensions on the stabilized mass flux is secondary to the influence of permeability heterogeneity. In the revised manuscript, we have included a concise discussion on the differences between 2D and 3D simulations of GDC.

The following sentences will be added to the conclusion Section 4.1 . "We note that in a more realistic 3D scenario, the dissolution rate may be approximately 25% higher than that observed in 2D cases. However, this difference is relatively minor when compared to the significant variability in permeability commonly encountered in geologic media Wang et al. (2022)." The revisions will be incorporated into the subsequent version.

(2). *The authors clearly state the software used for the numerical experiments and provide open access to the code and data. The user guide is extremely valuable, as it includes all the numerical details of the solver. However, I believe that for full reproducibility of this specific work, it would benefit from an additional short note, which I may have overlooked.*

R: Thank you for the suggestion regarding the reproducibility of the work. We agree that for full reproducibility, it would be beneficial to provide an additional note to clarify specific implementation details. We are sorry for not giving a very clear description to reach our conclusions. In the following, we give a summary that provides any potential information for full clarity, ensuring that all steps in the numerical experiment are fully accessible to users. Given that the reader has access to this document, we will not make further modifications to the original manuscript. We will neither put this on the general user guide for the MRST simulator because flow simulations are only used in Steps 2 and 3 described below.

Step 1. Generating heterogeneous fields based on the parameter given in Table 4 in the article. Random fields were generated using the sequential Gaussian simulation method implemented in the SGSIM code of GSLIB. *Note: The fields are generated in a random manner and could differ from those used in the present work. However, this does not impact the statistical conclusions drawn in this study.*

Step 2. Measuring the equivalent vertical ($\kappa_z^e$) and horizontal ($\kappa_x^e$) permeability for each realization of the random fields. To estimate $\kappa_i^e$ ($i = x, y$), we neglect gravity and saturate the porous medium with only water. We then set the domain sides perpendicular to the $i$th direction as impermeable, and we impose a pressure decrement $|\Delta_i p|$ along the $i$th direction. $\kappa_i^e$ is estimated by the total volumetric flow $Q_i$ passing through the system in the ith direction as $\kappa_i^e = \mu Q_i L_i / (A_i |\Delta_i p|)$, where $L_i$ is the domain size along the $i$th direction and $A_i$ the corresponding cross-sectional area. *Note: To maintain the simplicity and clarity of the code library, we have omitted this specific calculation in the uploaded code, [https://zenodo.org/records/5833962](https://zenodo.org/records/5833962). However, the calculation can be readily implemented by making minor modifications to the code provided in our uploaded file. Alternatively, it can also be achieved using existing functionalities in the MRST [https://www.sintef.no/projectweb/mrst/](https://www.sintef.no/projectweb/mrst/).*

Step 3. Conducting GDC (Gravity-Driven Convection) simulations with the provided code (located in *benchmarks and examples/example_unstable_finger*). It is essential to capture two key pieces of information during the simulation process: the mass flux through the top boundary, denoted as $F(t)$, and the detailed flow field. *Note: The example code is designed to be broadly applicable. Users are required to tailor the simulation parameters—discretization, domain size, permeability, and porosity—according to the specifications outlined in Table 3. The methodology for calculating $F(t)$ is as follows:*

$F(t) = \frac{1}{L} \int_0^L \left[ \rho X^C q_z - \phi D \rho \frac{\partial X^C}{\partial z} \right]_{z=0} dx$ (c.f. Equation (15) in the manuscript)

*The flow field is used to calculate the finger velocity $v(t)$, which is given as follows:*

$v(t) = \max_{0 < z < B} \left\{ \frac{1}{L} \int_0^L \frac{1}{\phi} |q_z| dx \right\}$ (c.f. Equation (16) in the manuscript)

Step 4. Estimating the Asymptotic Values of the Dissolution Rate ($F_\infty$) and Vertical Finger Velocity ($v_\infty$). In our simulations, the asymptotic values of the global dissolution rate ($F_\infty$) and the vertical finger velocity ($v_\infty$) are determined by computing the temporal averages of $F(t)$ and $v(t)$ over the interval $[\frac{t_b}{3}, t_b]$. Here, $t_b$ denotes the time at which the earliest aqueous $CO_2$ finger reaches the bottom, marked by the moment when the maximum $CO_2$ mass fraction at the bottom exceeds 25% of the $CO_2$ concentration at the top boundary.

Step 5. Obtaining the predictors for the asymptotic dissolution rate by performing regression analysis for the simulation results (summarized in https://zenodo.org/records/14061632) based on

$F_\infty^* = \gamma_1 \left( \frac{\kappa_z^e}{\kappa_g} \right)^{\alpha_1} \left( \frac{\kappa_x^e}{\kappa_z^e} \right)^{\beta_1}$ (c.f. Equation (17) in the manuscript)

and

$F_\infty^* = \gamma_1 \left( \frac{v_\infty}{v_c} \right)^{\alpha_2} \left( \frac{\kappa_x^e}{\kappa_z^e} \right)^{\beta_2}$ (c.f. Equation (18) in the manuscript)

Finally, we will get the results listed in Table 5 of the manuscript.

(3) *Multiple realizations of permeability distributions are provided, and Figure 4 effectively illustrates the differences in numerical simulation results between them. I was wondering whether the upscaling of highly heterogeneous permeability fields could yield similar results for the predictors or if a relationship could be established between the upscaled and original permeability fields.*

R: This is a very good point. The question of upscaling permeability fields and its impact on our findings is an important one. The upscaling of heterogeneous permeability fields should statistically yield similar results as the original permeability fields, although the shapes of concentration profiles are different in original and upscaled fields.

**Proof by by Elenius and Gasda (2013):** The dissolution coefficient $\gamma$ obtained in heterogeneous fields by Elenius and Gasda (2013) is very similar to that in homogeneous field using equivalent permeability. This means that gravity-driven convection is governed by a rule similar to Darcy's law, with the density difference acting as the driving force, this relationship can be illustrated by the following equation: (cf. Equation (1) in the manuscript):

$$F_\infty = \gamma X_0^C \rho_0 \frac{\Delta \rho g \kappa}{\bar{\mu}} = \gamma X_0^C \rho_0 \frac{\kappa}{\bar{\mu}} \Delta \rho g \qquad (R1)$$

This means that the dissolution rate is statistically the same for heterogeneous fields and corresponding upscaled homogeneous fields of equivalent permeability.

**Proof from current work:** We first perform 10 GDC simulations with different random noise in a homogeneous field (or upscaled field) with permeability $\kappa = 10^{-12}$ m², and then perform GDC simulations in different isotropically heterogeneous fields with geometric mean permeability $\kappa_g = 10^{-12}$ m², we obtain that the relation of the dissolution rate follows $\frac{F_{hetero}}{F_{homo}^{mean}} = \frac{\kappa_z^e}{\kappa}$, where $\kappa_z^e$ is the equivalent permeability, $F_{homo}^{mean}$ and $F_{hetero}$ are the statistic dissolution rate in homogeneous field (or upscaled field) and the dissolution rate in the heterogeneous field. This means the dissolution rate in the heterogeneous field can be obtained using an upscaled homogeneous field of equivalent permeability.

For anisotropic fields, we did not perform simulations in upscaled anisotropically homogeneous fields, but we can expect that upscaling should also work for anisotropic fields, because our predictors $F_\infty^* = \gamma_1 \left(\frac{\kappa_z^e}{\kappa_g}\right)^{\alpha_1} \left(\frac{\kappa_x^e}{\kappa_z^e}\right)^{\beta_1}$ (c.f. Equation (17) in the manuscript) and $F_\infty^* = \gamma_1 \left(\frac{v_\infty}{v_c}\right)^{\alpha_2} \left(\frac{\kappa_x^e}{\kappa_z^e}\right)^{\beta_2}$ (c.f. Equation (18) in the manuscript), which are based on equivalent permeabilities, perform well for all anisotropy, as shown in Figure R1 (c.f. Figure 7 in the manuscript):

[Figure]

Figure R1. Performance of predictors in fields of different equivalent vertical and horizontal

permeabilities $\kappa_z^e$ and $\kappa_x^e$.

Inspired by your comments, we add the following sentences in the manuscript:
"The results indicate that employing an upscaled permeability field with equivalent permeability does not compromise the depiction of dissolution efficiency in GDC simulations, although permeability upscaling does alter the shapes of the dissolution profiles." The revisions will be incorporated into the subsequent version.

**Minor Comments:**

(1). *Table 3: Could you comment on the potential effects of introducing a relationship between porosity and permeability realizations? Specifically, how might such a relationship influence the dissolution rates.*

R: This is indeed an insightful point. Introducing a relationship between porosity and permeability to account for heterogeneous porosity distribution may subtly influence the morphology of instability fingers, because the pore size may affect the interstitial velocity. However, it does not statistically impact the dissolution rates. This is because gravity-driven convection is governed by a rule similar to Darcy's law, as illustrated by the equation (R1), which clearly shows that the dissolution rate depends solely on the equivalent permeability and is independent of porosity.

We further explan the use of constant porosity in our simulations for two additional reasons:
(i)Limited Porosity Variation: The range of porosity variation (0.1–0.38) is relatively narrow compared to the wide range of intrinsic permeability ($\kappa = 10^{-16} - 10^{-12}$ m$^2$ and $\sigma_{\ln \kappa}^2$ = 3-10), as documented in Table 3 of Wang et al. (2022) and Table 4 of Elenius and Johannsen (2012).(ii)Ambiguity in Permeability-Porosity Relationships: On one hand, clay particles are significantly smaller than sand particles, resulting in a higher total pore space in clay soils. However, these pores are typically small and poorly connected, leading to low permeability. In contrast, sand particles are larger and more irregularly shaped, creating larger and better-connected pores that facilitate higher permeability. Thus, permeability is influenced not only by pore volume but also by pore shape, meaning that high porosity does not necessarily imply high permeability. On the other hand, for a given aquifer, increasing porosity through acid water erosion often leads to an increase in permeability, as described by the well-known Kozeny-Carman model (Saaltink et al., 2013). However, even if we employ the Kozeny-Carman model to represent the permeability-porosity relationship, the model parameters are typically site-specific, limiting their generalizability.

(2). *Line 231: 553 realizations and line 354: 554 realizations.*

R: We apologize for the typo. We corrected this in the manuscript. The total number of simulations should be 554.

*(3). In Figure 4, the range of permeability values appears to be relatively narrow (from -14 to -10 on a logarithmic scale). Could you comment on the potential impact of using a wider range of permeability values?*

R: This is a very good point. We will respond your comments in two aspects.

(1) When the geometric mean permeability and correlation length are held constant while the variance ($\sigma_Y^2$) increases to produce a broader spectrum of permeability values, the influence of permeability heterogeneity becomes more pronounced, and the connectivity within the medium improves. Consequently, preferential flow tends to occur within the interconnected high-permeability zones. In this context, the uncertainty associated with the development of instability fingers is primarily governed by the permeability heterogeneity, while the role of white noise, which initially triggers the instability, becomes relatively minor. Essentially, the flow becomes focused in the high-permeability regions regardless of the specific initial perturbation.

**Note:** In a medium with minimal heterogeneity, instability fingers can emerge due to minor white noise present in the initial conditions. This can result in variations in finger shapes across different simulations using distinct white noise inputs. However, the statistical dissolution rate remains consistent, as demonstrated by Pau et al. (2010) and this work.

Irrespective of the behavior of $CO_2$-rich fingers, the overall vertical mass flux of $CO_2$ can be reliably predicted based on the equivalent vertical and horizontal permeabilities. These equivalent permeabilities can be calculated using the method detailed in Section 4.5 of the manuscript.

Therefore, we expect that employing a large variance ($\sigma_Y^2$) values amplifies preferential channeling within interconnected high-permeability zones and may consequently affect dissolution rates. Nonetheless, this observation does not undermine the conclusion that $CO_2$ dissolution rates can be reliably estimated using equivalent vertical and horizontal permeabilities. To put it succinctly, the fundamental relationship between dissolution rate and equivalent permeabilities remains consistent, regardless of the permeability variability.

(2) When the variance ($\sigma_Y^2$) is negligible while the average permeability changes, the size of the instability fingers is inversely proportional to the permeability, as described by the relation $l_c = 70 \cdot \frac{\mu \phi D_m}{\Delta \rho g \kappa_g}$. This relationship indicates that the finger size is very large in media with

very low permeability. This insight is particularly valuable when designing simulation domains or laboratory experiments, since an inadequately chosen domain size may fail to accurately represent gravity-driven convection phenomena.

For instance, consider a scenario where the characteristic length of the fingers ($l_c$) is 1 meter. In such cases, employing a simulation domain or experimental reservoir smaller than 1 meter may fail to accurately capture the development of instability fingers. Given a specific size of the experimental reservoir, meticulous selection of sand permeability becomes essential to ensure that the observed finger distributions are both representative and meaningful.

Therefore, we claim that changing the mean permeability in a field with negligible heterogeneity will change the finger size, and the size of the simulation domain should be changed accordingly to efficiently match the density instability fingers.

We add the following comments in the revised manuscript:
"From Figure 7, it is also evident that the performance of our predictors is not influenced by permeability. This suggests that the findings of this study can be extended to fields with greater permeability heterogeneity." The revisions will be incorporated into the subsequent version.

**Reference:**

Elenius, M. T. and Johannsen, K.: On the time scales of nonlinear instability in miscible displacement porous media flow, Computational Geosciences, https://doi.org/10.1007/s10596-012-9294-2, 2012.

Elenius, M. T. and Gasda, S. E.: Convective mixing in formations with horizontal barriers, Advances in Water Resources, https://doi.org/10.1016/j.advwatres.2013.10.010, 2013.

Pau, G. S., Bell, J. B., Pruess, K., Almgren, A. S., Lijewski, M. J., and Zhang, K.: High-resolution simulation and characterization of density-driven flow in CO2 storage in saline aquifers, Advances in Water Resources, 33, 443 – 455, https://doi.org/https://doi.org/10.1016/j.advwatres.2010.01.009, 2010

Saaltink, M. W., Vilarrasa, V., De Gaspari, F., Silva, O., Carrera, J., and Rötting, T. S.: A method for incorporating equilibrium chemical reactions into multiphase flow models for CO2 storage, Advances in Water Resources, 62, 431 – 441, https://doi.org/10.1016/j.advwatres.2013.09.013,2013.

Wang, Y., Fernàndez-Garcia, D., and Saaltink, M. W.: Carbon Dioxide (CO2) Dissolution Efficiency During Geological Carbon Sequestration (GCS) in Randomly Stratified Formations, Water Resources Research, 58, e2022WR032 325, https://doi.org/10.1029/2022WR032325, 2022.

---

## Author Comment (AC3)

Response to reviewer 3 (Dr. Giacomo Medici)

**General comments**

*Very good research on geological carbon storage. Please, take into account my comments to improve the manuscript.*

R: We appreciate your recognition of the quality of our research and your valuable suggestions for improving the manuscript. Below, we provide a detailed response to each of your comments and explain how we have addressed them in the revised manuscript. The following revisions will be incorporated into the subsequent version.

**Specific comments**

(1). *Lines 15-87. Please, more emphasis on the fact that your research has a double focus: anisotropy and heterogeneity. My suggestions aim to increase the impact of your research in different sub-fields of applied geoscience.*

R: Thank you. This is a good suggestion. We have revised the introductory section to explicitly state that the objective of this work is twofold. (i) To quantitatively analyze the effect of permeability heterogeneity and anisotropy on the GDC-driven dissolution rate in a wide range of (isotropic and anisotropic) heterogeneous fields with varying degrees of heterogeneity and anisotropy. (ii) To investigate whether the dissolution rate can be predicted based on the finger-tip velocity. Here, we compact the anisotropy and heterogeneity as one focus. We also emphasize how our research can be employed in different sub-fields of applied geoscience by introducing its potential application in other gravity-driven convection processes. We have modified the introduction as follows:

"Therefore, the objective of this work is twofold. (i) To quantitatively analyze the effect of permeability heterogeneity and anisotropy on the GDC-driven dissolution rate in a wide range of (isotropic and anisotropic) heterogeneous fields with varying degrees of heterogeneity and anisotropy. (ii) To investigate whether the dissolution rate can be predicted based on the finger-tip velocity. We do this in two steps. First, performing numerical simulations over a large number of heterogeneous fields of different permeability distributions. Numerical simulations are carried out by a finite-difference numerical program developed by Wang (2022). Permeability fields are generated with the sequential Gaussian simulation method implemented in the SGSIM code (Journel and Huijbregts, 1976). Second, the results of the simulations are analyzed to find relations among the GDC-driven dissolution rate, permeability heterogeneity, anisotropy and finger-tip velocity, and we compare our results against those given in literature. In this step, ordinary-least-squares linear regressions are used. The conclusions from this work may hold significant relevance for other gravity-driven convection processes, where density differences play a crucial role. These processes include contaminant migration, geothermal exploitation, saltwater intrusion, and mineral

precipitation/dissolution (Berhanu et al., 2021; Sanz et al., 2022; Guevara Morel and Graf, 2023; Fang et al., 2024; Liyanage et al., 2024)." The revisions will be incorporated into the subsequent version.

We also give a short summary of the shortage of current research before we introduce our objectives:

"Overall, we have a solid understanding of the GDC-driven dissolution process in isotropic homogeneous media, but the GDC-driven dissolution in heterogeneous media needs further study. Especially, we need to quantitatively clarify the impact of the anisotropy ratio on the effective dissolution rate. Moreover, the current predictors are all based on the (equivalent) permeability, and it remains unclear whether we can predict the dissolution rate based on other formation properties or field observations, such as the finger-tip velocity."

(2). *Lines 21-22. "By being sealed under the low permeability caprock". Consider inserting recent papers on low permeability sedimentary layers that incorporate a discussion on applications for CO2 storage.*

*- English, K.L., English, J.M., Moscardini, R., Haughton, P.D., Raine, R.J. and Cooper, M., 2024. Review of Triassic Sherwood Sandstone Group reservoirs of Ireland and Great Britain and their future role in geoenergy applications. Geoenergy, 2(1), pp.geoenergy2023-042.*

*- Medici, G., Munn, J.D., Parker, B.L., 2024. Delineating aquitard characteristics within a Silurian dolostone aquifer using high-density hydraulic head and fracture datasets. Hydrogeology Journal, 32(6), 1663-1691.*

*- Newell, A.J. and Shariatipour, S.M., 2016. Linking outcrop analogue with flow simulation to reduce uncertainty in sub-surface carbon capture and storage: an example from the Sherwood Sandstone Group of the Wessex Basin, UK. Geological Society, London, Special Publications, 436(1), 231-246.*

R: Thank you. We have read these interesting papers and added them in the reference list. You will see them into the subsequent version.

(3). *Line 87. Please, disclose the specific objectives of your research by using numbers (e.g., i, ii, and iii).*

R: Thank you. Inspired by your suggestions we disclose the specific objectives of our research by using numbers (i) and (ii), to clarify the focus of the article, as shown in the following:

"Therefore, the objective of this work is twofold. (i) To quantitatively analyze the effect of permeability heterogeneity and anisotropy on the GDC-driven dissolution rate in a wide range of (isotropic and anisotropic) heterogeneous fields with varying degrees of heterogeneity and

anisotropy. (ii) To investigate whether the dissolution rate can be predicted based on the finger-tip velocity." The revisions will be incorporated into the subsequent version.

(4). *Line 133. Be more specific when you mention anisotropy. "Flow anisotropy"? We are not talking about a mechanical anisotropy here due to the presence of rock discontinuities.*

R: This is a good suggestion. Your suggestions help ensure clarity and precision in the manuscript. Here, anisotropy refers to "permeability anisotropy", which specifically describes the directional dependence of permeability (or hydraulic conductivity). It is often used when discussing how permeability varies with direction in a porous medium. For example: "The aquifer exhibits strong permeability anisotropy, with horizontal permeability significantly higher than vertical permeability." It is a key concept in hydrogeology because it directly influences fluid flow patterns can may lead to *flow anisotropy*.

According to your suggestion. We have specified it in this sentence and other sentences using "anisotropy" in the manuscript when necessary. Please check in the upcoming revised version.

(5) Line 295. Too soon for introducing references in this paragraph.

R: Thank you for this suggestion. We have reorganized this paragraph and introduce it later in the paragraph. The changed version is as follows: "For illustrative purposes, we chose a representative permeability realization for each case. These realizations are shown in panel (a) of Figure 4, from which we can see that $CO_2$ fingering is strongly affected by heterogeneity. In particular, the presence of vertical well-connected high permeability zones (preferential channels) facilitates the initiation and growth of the instability fingers (see for instance the second column of Figure 4). Actually, the white randomness of the top $CO_2$ mass fraction (needed in homogeneous media to create instabilities) is redundant in heterogeneous porous media as instabilities are controlled by these vertical preferential channels. In all cases, results show that instability make $CO_2$ fingers grow, merge and re-initiate as also observed in laboratory experiments (Rasmusson et al., 2017; Liyanage, 2018; Tsinober et al., 2022) and numerical simulations (Elenius et al., 2015)." The revisions will be incorporated into the subsequent version.

(6) *Line 367. "Geological Carbon Sequestration (GCS)". You have already defined the acronym.*

R: Thank you. We have removed the redundant definition of GCS to avoid repetition. However, we have retained the definition of Gravity-Driven Convection (GDC) in this paragraph, despite its prior definition. By redefining GDC, we aim to immediately connect readers with its meaning, facilitating smoother comprehension. While acronyms streamline writing, they can occasionally hinder readability, especially in lengthy articles, as readers may need to backtrack to recall their definitions. To mitigate this inconvenience, we have chosen

to reintroduce the meaning of GDC at its first appearance in each section. We trust you understand the rationale behind this approach.

(7) *Line 424. Integrate the two relevant papers on the effect of low permeability layers on deep aquifers/reservoirs.*

R: We have carefully read and added English et al. (2024), Medici et al. (2024) and Newell & Shariatipour (2016) in the reference lists. Thank you for your recommendation. These additions provide a more comprehensive perspective on the topic.

**Figures and tables**

(8) Figure 1a. You made bigger part of the reservoir with a rectangle. The spatial scale is unclear.

R: This is a very good question. The size of instability fingers is a function of permeability, as described by $l_c = 70 \cdot \frac{\mu \phi D_m}{\Delta \rho g \kappa_g}$. This relationship underscores that the size of the instability fingers and the simulation domain are contingent upon the permeability. Given the substantial variability in permeability across different reservoirs, the dimensions of instability fingers can differ markedly. Consequently, it is challenging to assign a precise spatial scale to the general illustration of instability fingers.

Ultimately, Figures 1a and 1b are designed to convey general insights into the development of instability and vertical convection, thereby introducing the objectives of this work. As such, specific temporal and spatial scales are not provided. We trust that you understand this approach.

(9) *Figure 2. Spatial scale unclear also here.*

R: Thank you. We add the following descriptions in the caption instead of putting them inside the figure to make the sketch concise: "The size of the simulation domain is B × L=7.5×7.5 [m²], and other hydrogeology properties are summarized in Table 2." The revisions will be incorporated into the subsequent version.

(10). *Figure 4. Make the letters larger.*

R: Thank you. In Figure 4, the font size of letters and numbers has been increased to enhance readability. A figure with larger letters and numbers will be found in the subsequent version.

(11). *Figure 4. Make the letters and numbers larger.*

R: Thank you. Probably you mean Figure 5, because you have mentioned Figure 4 in the previous comment. We notice that the letters and numbers are not large enough in Figure 5,

and we increase the font size of all labels, letters, and numbers in Figure 5. A figure with larger letters and numbers will be found in the subsequent version.